# LONG-TAILED DIFFUSION MODELS WITH ORIENTED CALIBRATION

**Tianjiao Zhang**[1]**, Huangjie Zheng**[3]**, Jiangchao Yao**[1,2]**, Xiangfeng Wang**[4]**, Mingyuan Zhou**[3]**,
Ya Zhang**[1,2]**, Yanfeng Wang**[1,2,✉]
[1] Cooperative Medianet Innovation Center, Shanghai Jiao Tong University
[2] Shanghai Artificial Intelligence Laboratory
[3] The University of Texas at Austin
[4] East China Normal University
{xiaoeyuztj, Sunarker, ya_zhang, wangyanfeng}@sjtu.edu.cn,
xfwang@cs.ecnu.edu.cn, huangjie.zheng@utexas.edu,
mingyuan.zhou@mccombs.utexas.edu

## ABSTRACT

Diffusion models are acclaimed for generating high-quality and diverse images. However, their performance notably degrades when trained on data with a long-tailed distribution. For long tail diffusion model generation, current works focus on the calibration and enhancement of the tail generation with head-tail knowledge transfer. The transfer process relies on the abundant diversity derived from the head class and, more significantly, the condition capacity of the model prediction. However, the dependency on the conditional model prediction to realize the knowledge transfer might exhibit bias during training, leading to unsatisfactory generation results and lack of robustness. Utilizing a Bayesian framework, we develop a weighted denoising score-matching technique for knowledge transfer directly from head to tail classes. Additionally, we incorporate a gating mechanism in the knowledge transfer process. We provide statistical analysis to validate this methodology, revealing that the effectiveness of such knowledge transfer depends on both label distribution and sample similarity, providing the insight to consider sample similarity when re-balancing the label proportion in training. We extensively evaluate our approach with experiments on multiple benchmark datasets, demonstrating its effectiveness and superior performance compared to existing methods. Code: https://github.com/MediaBrain-SJTU/OC_LT.

## 1 INTRODUCTION

Diffusion models have emerged as a powerful class of deep probabilistic models. These models leverage techniques from statistical physics and probabilistic modeling to generate high-quality, realistic samples from complex data distributions (Sohl-Dickstein et al., 2015). The effective implementation of a diffusion model necessitates extensive training on a diverse and sizable collection of image data. In general, there is a prevalent occurrence of a long-tail distribution (Yang et al., 2022), wherein a vast majority of images belong to a few dominant categories, while a significant portion of the dataset comprises less frequently occurring categories. As a consequence, the training of diffusion models with long-tail data continues to pose a formidable challenge owing to the distortion in the entire dataset.

In current works, many attempts have been made to address the lack of diversity and mode collapse issues in the generation of tail classes. A series of methods based on Generative Adversarial Networks (GANs) has been proposed. One of the common approaches is to adopt strategies that refine the general model's generation ability by improving its conditional modeling (Rangwani et al., 2022) on tail categories, which heavily depend on GANs structure. Another line of research focuses on alleviating the scarcity of samples in tail classes through appropriate data augmentation techniques (Karras et al., 2020; Zhao et al., 2020; Rangwani et al., 2023), and diffusion process (Zheng et al., 2023b; Wang et al., 2023). Nevertheless, such methods may not effectively capture the under-

lying data distribution or introduce meaningful variations (Yoo et al., 2020). For long tail diffusion models, Class Balancing Diffusion Models (CBDM) (Qin et al., 2023) has proposed a distribution adjustment regularizer enhancing tail generation based on the model prediction on the head class. However, the augmentation relying on the condition and prediction of the model might cause bias during training resulting in generated outcomes that do not meet expectations and leading to a lack of robustness.

In order to alleviate the issue, a direct knowledge transfer from head to tail categories should be established. Let's review a recent study on the diffusion process, Xu et al. (2023) observed that the score function exhibits the highest variance during the intermediate steps, which is a critical period for semantic formation (Zhang et al., 2023) with the undetermined target. Therefore, based on this evidence, we can utilize the score information from head classes to calibrate and enhance the generation of tail classes in this period and finally improve the overall generation performance. In this study, leveraging the evidence that the score of the diffusion model could be estimated via referencing multiple targets in the dataset (Xu et al., 2023), we propose a calibration strategy for the scores of the tail class directly making use of the head samples as reference, as shown in Figure. 1. By employing the strategy that leverages the similarity of underlying data distribution, the reliance on the conditional capacity of the model is mitigated and the generation performance is improved.

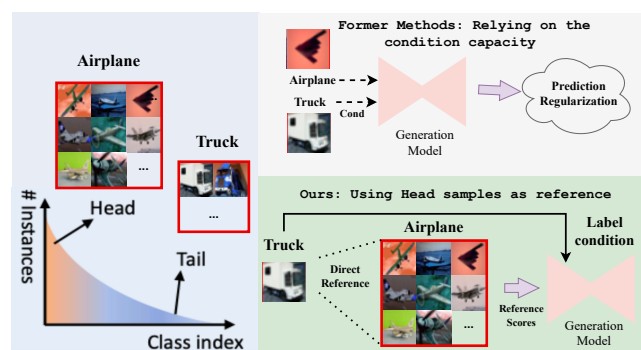

Figure 1: The illustration of our motivation. We directly utilize head samples as reference for tail augmentation instead of depending on the model condition capacity.

To realize the augmentation for the tail scores, we begin with modeling the score as the weighted scores averaging towards different targets. For conditional generation, the score estimation of noisy tail samples is augmented via properly improving the contribution of the score from the head samples in the reference batch, denoted as a T2H (noisy Tail to clean Head) operation to enhance the diversity of the tail generation. Simultaneously, a Batch Re-sample approach is utilized to alleviate the limitation on overall head-to-tail transfer strength in T2H mode. Besides that, in unconditional generation, the score function is predominantly influenced by samples from the head classes. Batch re-sampling is also employed to address this issue. While the method H2T (noisy Head to clean Tail), the reverse direction to T2H, has shown its effectiveness in unconditional generation.

Our contributions can be summarized as follows: (1) We have developed a method denoted as T2H based on the multi-target nature of score estimation to effectively calibrate and enhance the generation of tail classes in the semantic formation period, thereby significantly improving the overall generation performance. (2) A "Batch Re-sample" strategy is employed to construct a balanced reference batch, in order to address the extreme dominance issue of scores from the head class and promotes head-to-tail transfer under T2H mode. (3) We conduct extensive testing on three different long tail datasets (CIFAR10, CIFAR100, TinyImageNet) to validate the effectiveness of our proposed method. The results consistently demonstrated the superiority of our approach.

## 2 PRELIMINARY

The diffusion model involves slowly adding noise to the existing training data in the forward process, and then utilizing a deep learning network to gradually recover the original data from the noise in the reverse process. In the forward process, a clean image is progressively transformed by adding carefully calibrated Gaussian noise (Ho et al., 2020) at each diffusion step $t$, $q(x_t|x) = \mathcal{N}(\alpha_t x, \sigma_t^2 I)$, $0 \leq t \leq T$ where the coefficients $\alpha_t$ and $\sigma_t$ are chosen so that $q(x_t)$ is close to initial data density at $t \approx 0$ and close to Gaussian at $t \approx T$. For the reverse process, the diffusion model firstly samples from a Gaussian noise distribution $p(x_T) \sim \mathcal{N}(0, I)$, and then gradually incorporates

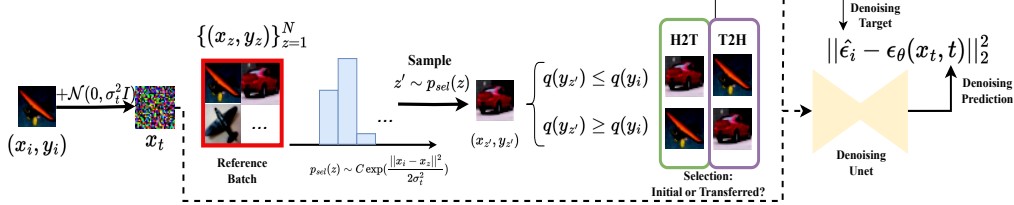

Figure 2: Overall flowchart for two strategies (H2T, T2H), where the target is determined by evaluating the probabilities of labels before and after the transfer to identify the satisfied mode.

various structures and semantic information at each step. Then transition $p(x_{t-1}|x_t)$ is estimated by training an image-to-image Unet parameterized by $\theta$, $p_\theta(x_{t-1}|x_t) = \mathcal{N}(x_{t-1}; \mu_\theta(x_t, t), \Sigma_t)$. Instead of directly predicting the mean value, the Unet is trained to predict the noise content of $x_t$, which is optimized by the following loss:

$$\mathcal{L}_{\text{Diff}}(\theta, x) = \mathbb{E}_{t \sim \mathcal{U}(0,T), \epsilon \sim \mathcal{N}(0,I)}[||\epsilon - \epsilon_\theta(x_t; t)||_2^2] \tag{1}$$

Another perspective on the diffusion model is to start from a parameter-free designed stochastic differential equation (SDE) $dx = f(x, t)dt + g(t)dw$, which transfers the initial data distribution to a prior distribution as time goes from 0 to $T$. The w is the standard Wigner process. Sampling the diffusion model via a reverse time SDE (Anderson, 1982): $dx = [f(x, t) - g(t)^2 \nabla_{x_t} \log p(x_t)]d\bar{t} + g(t)d\bar{w}$. The $\bar{\cdot}$ means time reverse. The score function namely $s(x_t, t) = \nabla_{x_t} \log p(x_t)$ should be estimated in the training stage. The predicted noise in Eq. (1) could be related to the score function via denoising score matching (Vincent, 2011):

$$\epsilon_\theta(x_t, t) = -\sigma_t s_\theta(x_t, t) \tag{2}$$

From Eq. (2) we could learn that the score is proportional to the noise predicted by the model. In this study, we start from the optimal score $\nabla_{x_t} \log p(x_t)$ to formulate our methodology for long tail distribution.

## 3 METHOD

Our problem involves training data that exhibits a long-tail distribution $(x^{(i)}, y^{(i)})_{i=1}^M$ sampled from $p(x, y)$ and a diffusion model parametrized via a denoising Unet $\epsilon_\theta(x_t, t)$. The $y^{(i)} \in \{C_1, C_2, ..., C_L\}$ is the label of $x^{(i)}$, assuming that the classes are ordered in descending probability of label occurrence, i.e., if $i < j$ then $n_i > n_j$, where the $n_i$ is the number training samples belonging to class $C_i$. With proper training methodology using long-tailed training data and diffusion model, we want to generate a more balanced and diversified data distribution $p^\star(x_0)$ in the inference time.

During the inference stage, a diffusion model utilizes a step-by-step reverse operation from a prior distribution $p(x_T)$ to data distribution $p(x_0)$ with reverse-SDE discussed in Section. 2. The reverse-SDE uses the score function $s(x_t, t)$ at time step $t$ obtained from the training stage. The optimal score $s^\star(x_t, t) = \nabla_{x_t} \log q_t(x_t)$ could be expressed with the expectation:

$$\nabla_{x_t} \log q_t(x_t) = \mathbb{E}_{q(x_0|x_t)} \nabla_{x_t} \log q(x_t|x_0), \tag{3}$$

where the proof is provided in Appendix A. As the equation illustrated, the score for a given $x_t$ could be calculated by the expectation under the distribution $q(x_0|x_t)$. Since this distribution is intractable, we utilize importance sampling to sample from initial data distribution $q(x_0)$.

Using the Bayesian rule, we have $q(x_0|x_t) = q(x_0)q(x_t|x_0)/q(x_t)$. Given $q(x_t) = \int q(x_t|x_0)q(x_0)dx_0$, we can derive

$$\mathbb{E}_{q(x_0|x_t)} \nabla_{x_t} \log q(x_t|x_0) = \mathbb{E}_{q(x_0)} \frac{q(x_t|x_0)}{\mathbb{E}_{x_0' \sim q(x_0)} q(x_t|x_0')} \nabla_{x_t} \log q(x_t|x_0). \tag{4}$$

Thus, with a mini-batch of samples from $q(x_0)$, i.e., $x_0^{(1:N)} \overset{iid}{\sim} q(x_0)$, we have

$$\mathbb{E}_{q(x_0|x_t)} \nabla_{x_t} \log q(x_t|x_0) \approx \sum_{i=1}^N \frac{q(x_t|x_0^{(i)})}{\sum_{j=1}^N q(x_t|x_0^{(j)})} \nabla_{x_t} \log q(x_t|x_0^{(i)}). \tag{5}$$

Specifically, we have expanded upon the original one-to-one optimal noise estimator in denoising score matching and transformed it into the one-to-many distributional matching technique, with a conditional distribution that determines the mapping probability, which encourages a mode-covering behavior in the score matching to enhance the diversity of generative modeling (Zheng & Zhou, 2021). The expectation in the equation we approximate with a weighted average of scores with respect to different samples $\{x_0^{(i)}\}$ denoted as a reference batch.

**T2H.** In conditional generation, the labels are involved and conditioned. The unconditional score $\nabla_{x_t} \log q(x_t)$ and conditional score $\nabla_{x_t} \log q(x_t|y)$ are both estimated in the training stage. With label participated, the sampling distribution should be $x_0^{(i)} \sim q(x_0|y)$ with label $y$ instead of $x_0^{(i)} \sim q(x_0)$ in Eq. (5) as:

$$\nabla_{x_t} \log q(x_t|y) \approx \sum_{i=1}^{N} \underbrace{\frac{q(x_t|x_0^{(i)}, y)}{\sum_j q(x_t|x_0^{(j)}, y_0^{(j)})}}_{\text{Score Weight}} \nabla_{x_t} \log q(x_t|x_0^{(i)}, y) \quad \{(x_0^{(k)}, y_0^{(k)})\}_{k=1}^{M} \sim q(x_0, y_0),$$

(6)

where the proof is provided in Appendix. B. In the denominator, we sample from $q(x_0, y_0)$ instead of $q(x_0)$ in the self-normalizing technique with label. The distribution of $q(x_t|x_0^{(i)})$ follows a Gaussian distribution and could be calculated by $B \exp(-\frac{||x_0^{(i)} - x_t||^2}{2\sigma_t^2})$ where $B$ is a normalizing constant and there is no label involved. The mixing weight $\frac{q(x_t|x_0^{(i)}, y)}{\sum_j q(x_t|x_0^{(j)}, y_0^{(j)})}$ from $(x_0^{(j)}, y_0^{(j)})$ is dominant by the $L_2$ distance respect to $x_t$. In addition to the $L_2$ distance in the Gaussian kernel, here we make a further assumption that the distribution $q(x_t|x_0, y_0)$ is adjusted by $q(y_0)^\beta$:

$$q(x_t|x_0, y_0) \propto B \, q(y_0)^\beta \, \exp(-\frac{||x_t - x_0||_2^2}{2\sigma_t^2}),$$

(7)

where $\beta$ is a pre-defined parameter, that controls the overall distribution density with respect to $y_0$. When $\beta = 0$ then the $x_t$'s distribution is only dependent on $x_0$.

For the score estimation of $x_t$ obtained from noisy tail sample $(x_0^T, y_0^T)$, we employ a method of score-oriented calibration. The approach enhances the contribution of head class samples $(x_0^H, y_0^H)$ by increasing the mixing score weight in Eq. 6. Consequently, it leverages the rich diversity of the head class to improve the performance of the tail class.

To improve the mixing score weight for the head samples, we could assign $\beta = 1$ in Eq. 7, since the $q(y_0^H)$ in the equation has a relative larger value. As shown in Figure. 3, the mixing weights towards head samples are improved compared with directly calculating the mixing weight depending only on $\exp(-\frac{||x_0^{(i)} - x_t||^2}{2\sigma_t^2})$.

Furthermore, for faster and easier implementation, we firstly sample a mini-batch $\{x_0^{(i)}, y_0^{(i)}\}_{i=1}^{K}$. Then as the typical training strategy of the diffusion model, for each sample $x_0^{(i)}$, we sample a random $t \sim \mathcal{U}(0, T)$ and random Gaussian noise $\epsilon_i \sim \mathcal{N}(0, I)$ and obtain the perturbed noisy $x_t$. The training objective is to predict the noise $\hat{\epsilon}_i$ by our parametrized Unet $\epsilon_\theta(x_t, t; y) = -\sigma_t s_\theta(x_t, t; y)$.

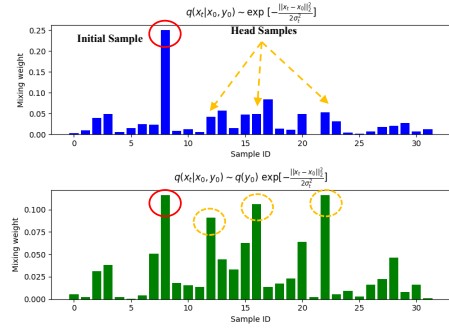

Figure 3: Mixing score weights of the sample with ID 8 towards different target in a reference batch under different weight calculation of $q(x_t|x_0, y_0)$.

Since the score for each target $x_t$, the $\nabla_{x_t} \log q(x_t|x_0^{(i)}, y) \propto -\frac{1}{\sigma_t}\epsilon_i$. As a consequence, we transfer the Eq. 6 from a score-weighted mixing problem to a score selection problem as:

$$\hat{\epsilon}_i = \epsilon_z; \quad z \sim p_{sel}(z) = \frac{q(x_t|x_0^{(z)}, y_0^{(z)})}{\sum_j q(x_t|x_0^{(j)}, y_0^{(j)})}; \quad 0 \leq z \leq K, \tag{8}$$

where $p_{sel}(z)$ denotes the probability of selecting score with index $z$ in the batch with size $K$. $\epsilon_z$ is calculated with $\epsilon_z = \frac{x_t}{\sqrt{\bar{\alpha}}} - x_0^{(z)}$, where the estimated noise is obtained by a new clean target and current noisy sample. So we firstly calculate the multi-nomial distribution in Eq. (8) with Gaussian kernel: $q(x_t|x_0^{(z)}, y_0^{(z)}) \propto B \exp(-\frac{||x_0^{(z)}-x_t||_2^2}{2\sigma_t^2})$, and then we sample z from this distribution. If $q(y_0^{(z)}) \geq q(y_0^{(i)})$, the transferring is allowed and the score is substituted with $\epsilon_z$, which means the noisy tail sample is mapped to a clean head sample. If not, the transferring is forbidden then the score is sent back to $\epsilon_i$, which means that the noisy head sample is forbidden to map to the clean tail sample. Since the noisy sample is obtained from the head class, we denote such kind of transferring mode as **T2H** (noisy Tail sample to clean Head sample).

Here, we amplify the contribution of head samples by selectively transferring the target solely to the clean head samples in the reference batch. The equivalence between T2H and directly enhancing the contribution of head samples via setting $\beta = 1$ in score mixing has also been validated in the experiments, as shown in Figure. 5.

We summarize the Algorithm in Alg. 1. We note for H2T mode, we just need to change the boundary condition from $q(y_0^{(z)}) \geq q(y_0^{(i)})$ in the algorithm to $q(v) \leq q(y_0^{(i)})$.

---

**Algorithm 1** T2H algorithm for conditional long tail generation

---

    Sample mini-batch $\{(x_0^{(i)}, y_0^{(i)})\}_{i=1}^K$ with balanced distribution $q^\star(x, y)$
    **for** each sample $(x_0^{(i)}, y_0^{(i)})$ in the mini-batch **do**
        Sample $x_t$ with random $t$ and Gaussian noise $\epsilon_i \sim \mathcal{N}(0, \sigma_t^2 I)$
        Calculate $p_{sel}(z)$ according to Eq. 8 with Gaussian kernel $C \exp(-\frac{||x_0^{(z)}-x_t||_2^2}{2\sigma_t^2}) \, 0 \leq z \leq K$
        Sample $z \sim p_{sel}(z)$
        **if** $q(y_0^{(z)}) \geq q(y_0^{(i)}) \, (q(y_0^{(z)}) < q(y_0^{(i)}))$ **then**
            $\hat{\epsilon}_i = \epsilon_z \, (\hat{\epsilon}_i = \epsilon_i)$
        **end if**
        Compute denoising loss
        **if** Conditional **then**
            $\mathcal{L}_{Diff} = ||\hat{\epsilon}_i - \epsilon_\theta(x_t, t; y)||_2^2$
        **else**
            $\mathcal{L}_{Diff} = ||\hat{\epsilon}_i - \epsilon_\theta(x_t, t)||_2^2$
        **end if**
    **end for**

---

For a specific sample of tail class $C_T$, we could augment the score via encouraging the model to predict the score towards the head class. Indeed, by leveraging the rich semantics of the head class and increasing the diversity of scores from the tail categories along the generation path, our approach enhances the overall diversity of generated samples. The increased variety ensures a more comprehensive generation of different classes and ultimately enhances the overall quality and diversity of the generated samples. The fidelity of the transferring could be found in Appendix. G.

**Batch Re-sample.** T2H achieves enhancement in generating tail categories through head-to-tail transfer for conditional generation. Here we evaluate the strength of the transfer quantitatively. Consider two samples $(x_0^H, y_0^H)$ and $(x_0^T, y_0^T)$ from different head class $C_H$ and tail class $C_T$. Let's discuss the distribution with same perturbed noise level $\mathcal{N}(0, \sigma_t^2 I)$ with $q(x_t|x_0^H, y_0^H)$ and $q(x_t|x_0^T, y_0^T)$. For the purpose of measuring the strength of transition from $C_T$ to $C_H$ based on Langevin dynamics, then the $\mathbb{E}_{q(x_t|x^H, y^H)} \frac{q(x_t|x_0^T, y_0^T)}{\sum_k q(x_t|x_0^{(k)}, y_0^{(k)})}$ should be calculated (Song & Ermon, 2020). Then we have the following proposition:

**Proposition 3.1** *Let $q(x_t|x_0, y_0) \propto B \ q(y_0)^\beta \ \exp(-\frac{||x_t - x_0||_2^2}{2\sigma_t^2})$, then:*

$$\mathbb{E}_{q(x_t|x_0^H, y_0^H)} \frac{q(x_t|x_0^T, y_0^T)}{\sum_k q(x_t|x_0^{(k)}, y_0^{(k)})} \leq \frac{B}{2}(q(y^H) \ q(y^T))^\beta \exp(-\frac{||x_0^H - x_0^T||_2^2}{8\sigma_t^2}). \tag{9}$$

As you can see from the proposition, the transition strength is determined by the $L_2$ distance represented by a Gaussian kernel between two samples and the product of label probability powered by $\beta$. In the perspective of Eq. (7) with algorithm T2H, when $\beta = 1$, The transfer strength is constrained by the product of the $q(y)$ of the head and tail classes, which is relatively a small value. Therefore, employing a Batch Re-sampling strategy during training, which equalizes the appearance probability of each category, can significantly enhance the head-to-tail transfer.

Besides, in unconditional generation, the training data $q(x_0)$ exhibits a long-tail distribution during the training process. As a result, If we directly use the results from the score estimated in Eq. (5), then the generated $p(x_0)$ will also follow a long-tail distribution. The frequent occurrence of head samples will lead to unconditional scores being dominated by the score of head samples. To intuitively illustrate the necessity of using the Batch Re-sample, we employed a toy example to illustrate the phenomenon of head class dominance in long-tail unconditional generation, as shown in Figure. 4. The Batch-Re-sample could also alleviate this issue. More can be found in Appendix. J.

Here, we follow the common assumption in the long tail recognition: the balanced distribution and initial long tail distribution are related by sharing the same conditional probability $q(x|y) = q^\star(x|y)$ (Zhang et al., 2013).

**H2T, Full.** In the former analysis, in order to augment the tail class, we propose a method denoted as noisy Tail to clean Head (T2H). Conversely, there would exist a method of H2T denoting noisy Head to clean Tail. In H2T, the weight of score towards target tail samples is improved corresponding to smaller $\beta$ in Eq. (7), eg. $\beta = -1$, as the inverse value $q(y)$ for the tail class is larger than the head class. Consequently, for a noisy head sample, the contribution of the tail sample targets are enhanced. In addition to H2T and T2H, if we do not assess the probabilities of transferred labels, this mode is denoted as 'Full' (means allowing both transfer directions).

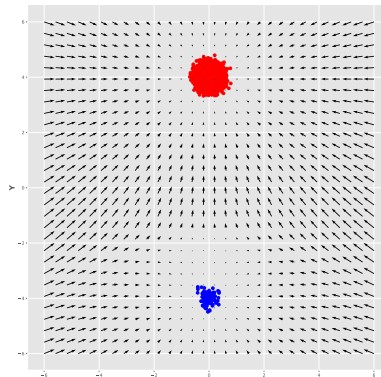

Figure 4: A toy example for simulating the score distribution. The red and blue points denote head and tail samples, respectively.

**Connection with CBDM (Qin et al., 2023).** The CBDM has employed a score aligning loss in the training stage: $\frac{1}{|\mathcal{Y}|}\sum_{y' \in \mathcal{Y}} t||\epsilon_\theta(x_t, y) - \epsilon_\theta(x_t, y')||^2$. The conditional score with label $y$ of $x_t$ is regularized with an another conditional score with label $y'$. The aligned strength is weighted with the diffusion time $t$. We could further prove that the optimal score of CBDM could be written as a weighted sum of the initial denoising score and an adjustment term, which is similar to Eq. (6). When the diffusion model converges, the optimal minimizer $\epsilon^*(x_t, y)$ for the CBDM loss: $\mathcal{L}_{CBDM}(x_t, y, t, \epsilon) = ||\epsilon_\theta(x_t, y) - \epsilon||^2 + \frac{\tau t}{|\mathcal{Y}|}\sum_{y' \in \mathcal{Y}} ||\epsilon_\theta(x_t, y) - \epsilon_\theta(x_t, y')||^2$ could be denoted as: $\epsilon^*(x_t, y) = \frac{1}{1+t\tau}\epsilon + \frac{t \tau}{(1+t \tau) |\mathcal{Y}|}\sum_{y'} \epsilon_\theta(x_t, y')$. where the proof is provided in Appendix. D. The operation also can be considered as augmenting the tail class with the head with labels. The difference is that the reference of our method is obtained from data, whereas CBDM's reference is based on the predictions and condition capacity of the model that may cause biased scores, as we have discussed in Section. 1. Our method possesses greater robustness besides the state-the-of-the-art performance.

## 4 EXPERIMENTS

**Experimental Setup.** We started by selecting two widely utilized datasets in the field of image synthesis, namely CIFAR10/CIFAR100, with their long-tailed versions CIFAR10LT and CIFAR100LT.

Table 1: Ablation study for both conditional and unconditional generation on CIFAR10LT.

| Model | Conditional | Batch Re-Sample | T2H | H2T | Full | FID ($\downarrow$) | IS ($\uparrow$) |
|-------|-------------|-----------------|-----|-----|------|--------------------|-----------------|
| A | | | | | $\checkmark$ | $25.31_{\pm0.12}$ | $7.01_{\pm0.02}$ |
| B | | $\checkmark$ | | | | $16.92_{\pm0.17}$ | $8.15_{\pm0.03}$ |
| C | | $\checkmark$ | | | $\checkmark$ | $16.85_{\pm0.10}$ | $8.16_{\pm0.02}$ |
| D | | $\checkmark$ | $\checkmark$ | | | $16.78_{\pm0.09}$ | $8.11_{\pm0.05}$ |
| E | | $\checkmark$ | | $\checkmark$ | | $16.09_{\pm0.11}$ | $8.27_{\pm0.02}$ |
| F | $\checkmark$ | | | | | $10.72_{\pm0.23}$ | $9.37_{\pm0.03}$ |
| G | $\checkmark$ | $\checkmark$ | | | | $10.20_{\pm0.13}$ | $9.65_{\pm0.01}$ |
| H | $\checkmark$ | $\checkmark$ | | $\checkmark$ | | $8.20_{\pm0.09}$ | $9.77_{\pm0.01}$ |
| I | $\checkmark$ | $\checkmark$ | | | $\checkmark$ | $7.52_{\pm0.12}$ | $9.73_{\pm0.02}$ |
| J | $\checkmark$ | $\checkmark$ | $\checkmark$ | | | $6.89_{\pm0.09}$ | $9.75_{\pm0.05}$ |

The construction of CIFAR10LT and CIFAR100LT follows the methodology proposed in Cao et al. (2019) , wherein the size of each category exponentially decreases with its category index, adhering to an imbalance factor of imb = 0.01. For the CIFAR10LT dataset, we also implement a more skewed version with the imbalance factor of imb = 0.001. Two commonly used metrics for image generation are adopted namely Frechet Inception Distance (FID) (Heusel et al., 2017) and Inception Scores (IS) (Salimans et al., 2016). During the inference time, we generate 50k images for the evaluation of the metrics. A DDIM sampler (Song et al., 2020a) is utilized with 100 steps of 10 steps skip comparison with initial DDPM (Ho et al., 2020) 1000 steps.

Our training schedules are strictly follows the implementation of CBDM (Qin et al., 2023), which follows the DDPM settings. The experiments are conducted with two settings corresponding to the methods, unconditional generation and conditional generation. For the unconditional generation, we only adjust the training strategy with no label information injected into the diffusion model for optimization. At the inference stage, the unconditional diffusion model is asked to generate 50k images freely. For the conditional generation, there is label information injected into the diffusion model in the training stage. While at the inference stage, the diffusion model is asked to generate $50\mathrm{k}/L$ images for each class where $L$ is the number of classes.

Table 2: Comparison with other long tail generation methods

| Datasets with different imb factors | CIFAR10LT | | | | CIFAR100LT | |
|-------------------------------------|-----------|----|----|----|------------|----|
| | Imb factor 0.01 | | Imb factor 0.001 | | Imb factor 0.01 | |
| # Metrics | FID ($\downarrow$) | IS ($\uparrow$) | FID ($\downarrow$) | IS ($\uparrow$) | FID ($\downarrow$) | IS ($\uparrow$) |
| CBGAN (Rangwani et al., 2021) | 37.23 | 6.01 | 46.61 | 5.77 | 33.01 | 7.04 |
| gSR-GAN (Rangwani et al., 2022) | 12.86 | 8.56 | 38.71 | 6.89 | 13.96 | 10.01 |
| DDPM (Ho et al., 2020) | 10.72 | 9.37 | 15.00 | 9.16 | 10.25 | **12.96** |
| CBDM (Qin et al., 2023) | 7.27 | 9.37 | 12.71 | 9.01 | 7.82 | 12.40 |
| Ours (DDPM+T2H) | **6.89** | **9.75** | **11.56** | **9.17** | **6.68** | 12.94 |

**Ablation study.** We conduct both conditional and unconditional generation experiment on CIFAR10LT, the strategies that have been discussed are applied to a base DDPM unconditional/conditional generation model. As shown in the Table 1, in an unconditional generation, the Batch Re-sample with H2T and T2H could improve the performance, where H2T is more effective as we discussed in the former section. In a conditional generation, T2H is more efficient than H2T and 'Full' since improving the contribution of head samples for the noisy tail sample could enhance the diversity of tail classes and promoting the overall generation performance.

**Comparison with other methods.** We conduct a conditional generation, on CIFAR10LT, CIFAR100LT dataset. And we do a comparison with two GAN-based long-tail generation methods CBGAN (Rangwani et al., 2021) and gSR-GAN (Rangwani et al., 2022) and one diffusion-based methods CBDM. The results are shown in Table 2.

Table 3: Results on TinyImageNet200LT datasets with diffusion baselines

| Method | FID | IS |
|--------|-----|-----|
| Base DDPM | 19.24 | **18.20** |
| CBDM | 18.07 | 18.01 |
| Ours + T2H | **17.81** | 18.12 |

As illustrated in the Table 2, in the comparative analysis of conditional generation, the T2H method achieved the best performance of 6.89 FID score, surpassing base DDPM models with 3.83 and CBDM with 0.38. In a more skewed version of CIFAR10LT with an imbalanced factor of 0.001, T2H has achieved more than 1.0 FID improvements compared with CBDM. Under the CIFAR100LT benchmark, our method has also achieved more than 1.0 FID than the CBDM. The results have shown that our method has obtained a further improvement in the diffusion model on the long-tailed distribution, illustrating a more effective calibration and augmentation from data distribution than from the model prediction.

We also conduct experiments on a dataset TinyImageNet200LT with more classes and higher resolution, which is the long tail version of TinyImageNet200 (Tavanaei, 2020). The metric is based on 10k generated images referenced with its validation set, as shown in Table. 3

**T2H and H2T relation with different $\beta$ value.** For the purpose of validating the assumption with Eq. (7), we directly utilize the formula to calculate $p_{sel}(z)$ instead of H2T and T2H. As shown in Figure. 5, in the case of unconditional generation, the generation performance improves as the $\beta$ increases. Conversely, in the case of conditional generation, the generation performance deteriorates as the $\beta$ increases.

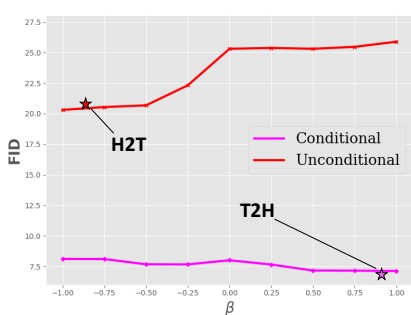

**Training Robustness of our method.** In order to achieve better performance for image generation, a diffusion model is required to train for a long time. Due to various forms of data imbalance, which is essentially the long-tail nature of the data, the model tends to overfit and consequently leads to a decrease in performance, as observed in Figure 6. This phenomenon is evident in both conditional and unconditional diffusion model training without any additional processing. However, when employing CBDM in conditional generation, this situation can be mitigated

Figure 5: FID scores versus different $\beta$ value. The performance of H2T (unconditional) and T2H (conditional) is also marked on the figure as pentagram.

to some extent. However, as training progresses, CBDM augmentation heavily relies on the model's prediction for other labels, so this phenomenon still exists. Nevertheless, our method effectively suppresses the issue and stabilizes performance in the long-time training process.

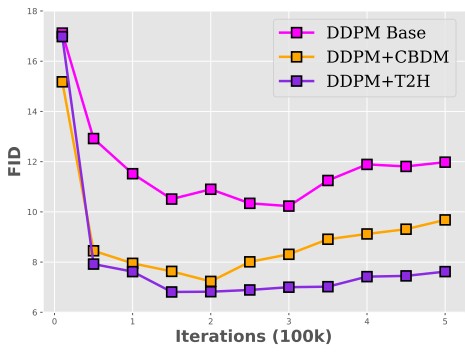

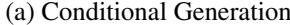

(a) Conditional Generation

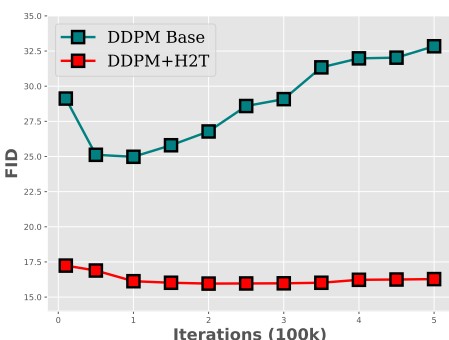

(b) Unconditional Generation

Figure 6: FID metric versus the long time training process on CIFAR10LT dataset. Our method could achieve better performance and stabilize the generation quality.

**Transfer Probability with diffusion time.** We make a study on selecting probability $p_{sel}(z)$ in Eq. (8) with diffusion time. The transfer probability $\sum_{z \neq i} p_{sel}(z)$, which denotes shifting the denoise target in the training stage, is counted and calculated with 100k samples perturbed with levels of noise corresponding to different diffusion times within a mini-batch. As shown in Figure. 7, when the diffusion step size is between 500 and 800, the probability of transferring the denoise target increases from 0 to 1 as the step size increases. If the step size exceeds 800, then the transfer will

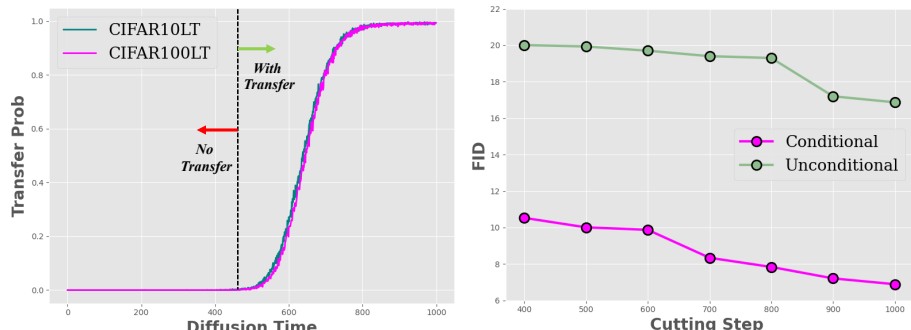

Figure 7: The left subfigure is the transfer probability versus diffusion time. The right subfigure shows the FID scores with different transfer cutting time.

occur with an approximately 1.0 probability. We also restrict the transfer cutting step, denoting that the target transfer (T2H, H2T) is only allowed below this diffusion step. The FID scores decrease dramatically for both conditional and unconditional generation with a cutting time range from 500-800, which is consistent with the observation of high variance phase in Xu et al. (2023).

## 5   RELATED WORK

**Diffusion Models.**   For the diffusion generation model, Ho et al. (2020) firstly announced that the training of the model is accomplished by utilizing a weighted variational bound. Song et al. (2020b) proposed an alternative approach to constructing a diffusion model, which involves utilizing a stochastic differential equation (SDE) that gradually injects noise to smoothly transform a complex data distribution into a known prior distribution. Karras et al. (2022) presents a design space that distinctly delineates the concrete design choices for former works. As for the diffusion process, Xu et al. (2023) has observed three phases with distinct behaviors affecting the generation diffusion model. Raya & Ambrogioni (2023) demonstrate that the diffusion process could be modeled in a manner analogous to symmetry breaking in physics.

**Long Tail Recognition.**   Long tail recognition refers to the task of accurately recognizing and classifying rare or infrequently occurring classes in a given dataset together with frequently occurring classes (Zhou et al., 2022). There are several approaches to address the problem, including re-weighting (Huang et al., 2016), logit adjustment (Menon et al., 2020; Zhou et al., 2023), robust distributional matching (Zheng et al., 2023a; Chen et al., 2024), and knowledge transfer (Wang et al., 2017; Chen et al., 2022; 2023b). Cui et al. (2019) declare that as the number of samples increases, the diminishing phenomenon suggests that there is a decreasing marginal benefit for a model to extract additional information from the data due to the presence of information overlap.

**Long Tail Generation.**   The objective of long tail generation is to generate a more balancing and diverse dataset training with a long tail dataset. CB-GAN (Rangwani et al., 2021) has used a regularizer that makes use of a pretrained classifier in the training stage to ascertain the balance learning of all classes in the dataset. gSR-GAN (Rangwani et al., 2022) observes that the performance decline in long tail generation primarily occurs because of class-specific mode collapse in tail classes which correlated with the spectral explosion of the conditioning parameter matrix and proposes a corresponding group spectral regularizer. CBDM (Qin et al., 2023) makes use of a distribution adjustment regularizer in the training stage for the purpose of augmenting the tail classes.

## 6   CONCLUSION

The main challenge for the long-tail diffusion generation is the lack of diversity for the tail class generation. To tackle the challenge, based on the multi-target characteristic of denoising score, a T2H augmentation for the estimation of noisy tail samples is achieved improving the score contribution of head samples in the reference batch. At the same time, the Batch Re-sample operation helps alleviate the dominant effect of head samples on the scores and promote head-to-tail transfer. The experiments are conducted to validate our approach on multiple benchmark datasets, demonstrating effectiveness compared with baseline methods and robustness versus training time.

ACKNOWLEDGEMENT

This work is supported by the National Key R&D Program of China (No. 2022ZD0160702), STCSM (No. 22511106101, No. 22511105700, No. 21DZ1100100), 111 plan (No. BP0719010) and National Natural Science Foundation of China (No. 62306178). We also thank Fei Zhang for his insightful discussion.

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

## A  THE EXPECTATION FORMULATION OF OPTIMAL SCORE FOR $x_t$.

The optimal score for $x_t$ could be calculated as :

$$\nabla_{x_t} \log q(x_t) = \mathbb{E}_{q(x_0|x_t)} \nabla_{x_t} \log q(x_t|x_0)$$

**Proof A.1** *Based on Bayes theorem and definition of expectation:*

$$
\begin{aligned}
\nabla_{x_t} \log q(x_t) &= \frac{1}{q(x_t)} \nabla_{x_t} q(x_t) \\
&= \frac{1}{q(x_t)} \nabla_{x_t} \int q(x_t|x_0) q(x_0) \mathrm{d}x_0 \\
&= \int \frac{q(x_0)}{q(x_t)} \nabla_{x_t} q(x_t|x_0) \mathrm{d}x_0 \\
&= \int \frac{q(x_t|x_0) \, q(x_0)}{q(x_t)} \nabla_{x_t} \log q(x_t|x_0) \mathrm{d}x_0 \\
&= \int \frac{q(x_t, x_0)}{q(x_t)} \nabla_{x_t} \log q(x_t|x_0) \mathrm{d}x_0 \\
&= \int q(x_0|x_t) \nabla_{x_t} \log q(x_t|x_0) \mathrm{d}x_0 \\
&= \mathbb{E}_{q(x_0|x_t)} \nabla_{x_t} \log q(x_t|x_0)
\end{aligned}
$$

## B  THE SCORE FOR THE CONDITIONAL GENERATION

The optimal score for $x_t$ given $y$ could be calculated as:

$$\nabla_{x_t} \log q(x_t|y) \approx \sum_{i=1}^{N} \frac{q(x_t|x_0^{(i)}, y)}{\sum_j q(x_t|x_0^{(j)}, y)} \nabla_{x_t} \log q(x_t|x_0^{(i)}, y); \quad (x_0^{(i)}, y_0^{(i)}) \sim q(x_0, y_0) \tag{10}$$

**Proof B.1**

$$\nabla_{x_t} \log q(x_t|y) = \mathbb{E}_{q(x_0|x_t, y)} \nabla_{x_t} \log q(x_t|x_0, y)$$

*We use importance sampling:*

$$\mathbb{E}_{q(x_0|x_t,y)} \nabla_{x_t} \log q(x_t|x_0, y) = \sum_{x_0 \sim q(x_0|y)} \frac{q(x_0|x_t, y)}{q(x_0|y)} \nabla_{x_t} \log q(x_t|x_0, y) \tag{11}$$

*since conditional probability formula:*

$$\frac{q(x_0|x_t, y)}{q(x_0|y)} = \frac{q(x_t|x_0, y) q(x_0|y)}{q(x_t|y) q(x_0|y)} = \frac{q(x_t|x_0, y)}{q(x_t|y)} \tag{12}$$

*Compared with unconditional generation, we here calculate $q(x_t|y)$ as $q(x_t|y) = \int q(x_t|x_0, y) q(x_0|y) dx_0$, we utilize Monte-Carlo sample as:*

$$q(x_t|y) = \sum_j q(x_t|x_0^{(j)}, y) \quad x_0^{(j)} \sim q(x_0|y) \tag{13}$$

*We substitute the Equation into the Eq. (12):*

$$\frac{1}{N} \sum_{x_0^{(i)} \sim q(x_0|y)} \frac{q(x_t|x_0^{(i)}, y)}{\sum_j q(x_t|x_0^{(j)}, y)} \nabla_{x_t} \log q(x_t|x_0^{(i)}, y) \tag{14}$$

*Here, we want to augment the distribution $q(x_0|y)$ for a larger generation diversity. Here we analyze the $(\tilde{x}_0^{(j)}, y)$ that could not be sampled during the score estimation stage in the Eq. 14. For fixed little threshold probability $p_s$:*

*i) for the small $q(x_t|\tilde{x}_0^{(j)}, y) < p_s$, the final scores almost are not affected by these samples.*

*ii) for the $q(x_t|\tilde{x}_0^{(j)}, y) \geq p_s$, then we make a assumption that, there exists $(x_0^{(j')}, y_0^{(j')})$ that:*

$$\nabla_{x_t} \log q(x_t|x_0^{(j')}, y_0^{(j')}) \approx \nabla_{x_t} \log q(x_t|\tilde{x}_0^{(j)}, y) \quad (x_0^{(j')}, y_0^{(j')}) \sim q(x_0, y_0). \qquad (15)$$

*We make a little justification of the assumption. Suppose $x_t$ is obtained by adding sampled noise with probability $p_1$ to $(x_0^1, y)$, then we consider another sample $(\tilde{x}_0^2, y) \sim q(x_0|y)$ with probability $q(x_t|\tilde{x}_0^2, y) \geq p_s$. Under the $q(x_0|x_t, y) \propto C \exp(-\frac{||x_0 - x_t||_2^2}{2\sigma_t^2})$. We could easily check the similarity between $x_1$ and $\tilde{x}_0^2$, $||x_1 - \tilde{x}_0^2||_2^2 \leq -2\sigma_t^2(\log p_1 + \log p_s)$ following the similar procedure in Proposition. G.1. $\tilde{x}_0^2$ lies in a ball with radius $r = -2\sigma_t^2(\log p_1 + \log p_s)$ center at $x_1$.*

*If there exists a $(x_0^{2'}, y_0^{2'}) \sim q(x_0, y_0)$ in the ball, which could be sampled during the training stage with a another label $y_0^{2'}$. We could approximately substitute $\tilde{x}_0^2$ with $x_0^{2'}$ for probability and score. The substitution error of probability could be bounded with the $|q(x_t|x_0^{2'}, y_0^{2'}) - q(x_t|\tilde{x}_0^2, y)| \leq \max(q(x_t|\tilde{x}_0^2, y), q(x_t|x_0^{2'}, y_0^{2'}))(\log p_1 + \frac{r}{2\sigma_t^2})$. And the substitution error of scores could be bounded with $||\nabla_{x_t} \log q(x_t|x_0^{2'}, y_0^{2'}) - \nabla_{x_t} \log q(x_t|\tilde{x}_0^2, y)||_2^2 \leq \frac{r}{\sigma_t}$. The both substitution errors are suppressed at larger timesteps with larger $\sigma_t$.*

*We broad the distribution of $q(x_0|y)$ to entire distribution $q(x_0)$ as reference:*

$$\nabla_{x_t} \log q(x_t|y) \approx \sum_{i=1}^{N} \frac{q(x_t|x_0^{(i)}, y)}{\sum_j q(x_t|x_0^{(j)}, y)} \nabla_{x_t} \log q(x_t|x_0^{(i)}, y); \quad (x_0^{(i)}, y_0^{(i)}) \sim q(x_0, y_0) \qquad (16)$$

Note that in practice we can evaluate the density with $q(x_t|x_0^{(j)})$ as the diffusion process is not related to the label. So we here abuse the notation a little with substituting $q(x_t|x_0^{(j)}, y)$ with $q(x_t|x_0^{(j)}, y_0^{(j)})$ to represent the correspondence between $x_0^{(j)}$ and $y_0^{(j)}$:

$$\nabla_{x_t} \log q(x_t|y) \approx \sum_{i=1}^{N} \frac{q(x_t|x_0^{(i)}, y)}{\sum_j q(x_t|x_0^{(j)}, y_0^{(j)})} \nabla_{x_t} \log q(x_t|x_0^{(i)}, y); \quad (x_0^{(i)}, y_0^{(i)}) \sim q(x_0, y_0) \qquad (17)$$

Note that the sampling is initially operated with label $y$ instead of $y_0^{(j)}$.

## C  THE PROOF FOR THE PROPOSITION 3.1

**Proposition C.1** *Let $q(x_t|x_0^{(i)}, y_0^{(i)}) \propto B \, q(y_0^{(i)})^\beta \, \exp(-\frac{||x_t - x_0^{(i)}||_2^2}{2\sigma_t^2})$, then:*

$$\mathbb{E}_{q(x_t|x_0^{(i)}, y_0^{(i)})} \frac{q(x_t|x_0^{(j)}, y_0^{(j)})}{\sum_k q(x_t|x_0^{(j)}, y_0^{(j)})} \leq \frac{B}{2}(q(y_0^{(i)}) \, q(y_0^{(j)}))^\beta \exp(-\frac{||x_0^{(i)} - x_0^{(j)}||_2^2}{8\sigma_t^2})$$

**Proof C.1**

$$\mathbb{E}_{q(x_t|x_0^{(i)}, y_0^{(i)})} \frac{q(x_t|x_0^{(j)}, y_0^{(j)})}{\sum_k q(x_t|x_0^{(j)}, y_0^{(j)})} = \int \frac{q(x_t|x_0^{(i)}, y_0^{(i)})q(x_t|x_0^{(j)}, y_0^{(j)})}{\sum_k q(x_t|x_0^{(j)}, y_0^{(j)})} \mathrm{d}x_t$$

*while the term in the integral of the right side could be transformed into:*

$$\frac{q(x_t|x_0^{(i)}, y_0^{(i)})q(x_t|x_0^{(j)}, y_0^{(j)})}{\sum_k q(x_t|x_0^{(j)}, y_0^{(j)})} \leq \int \frac{q(x_t|x_0^{(i)}, y_0^{(i)})q(x_t|x_0^{(j)}, y_0^{(j)})}{q(x_t|x_0^{(i)}, y_0^{(i)}) + q(x_t|x_0^{(j)}, y_0^{(j)})} = \frac{1}{2} \int \frac{2}{\frac{1}{q(x_t|x_0^{(i)}, y_0^{(i)})} + \frac{1}{q(x_t|x_0^{(j)}, y_0^{(j)})}}$$

$$\leq \frac{1}{2} \sqrt{q(x_t|x_0^{(i)}, y_0^{(i)})p(x_t|x_0^{(j)}, y_0^{(j)})}$$

*Here we substitute the assumption $q(x_t|x_0^{(i)}, y_0^{(i)}) \propto B \, q(y_0^{(i)})^\beta \, \exp(-\frac{||x_t - x_0^{(i)}||_2^2}{2\sigma_t^2})$ into the formula:*

$$\frac{1}{2}\sqrt{q(x_t|x_0^{(i)}, y_0^{(i)})q(x_t|x_0^{(j)}, y_0^{(j)})} = \frac{B}{2}\sqrt{q(y_0^{(i)})^\beta \, \exp(-\frac{||x_t - x_0^{(i)}||_2^2}{2\sigma_t^2} q(y_0^{(j)})^\beta \, \exp(-\frac{||x_t - x_0^{(j)}||_2^2}{2\sigma_t^2})}$$

$$= \frac{B}{2}(q(y_0^{(i)})q(y_0^{(j)}))^\beta \sqrt{\exp(-\frac{||x_t - x_0^{(i)}||_2^2 + ||x_t - x_0^{(j)}||_2^2}{2\sigma_t^2})}$$

*Here we take a twice look at the integral and exclude terms unrelated to $x_t$:*

$$\int \frac{B}{2}(q(y_0^{(i)})q(y_0^{(j)}))^\beta \sqrt{\exp(-\frac{||x_t - x_0^{(i)}||_2^2 + ||x_t - x_0^{(j)}||_2^2}{2\sigma_t^2})}\mathrm{d}x_t$$

$$= \frac{B}{2}(q(y_0^{(i)})q(y_0^{(j)}))^\beta \int \exp(-\frac{||x_t - x_0^{(i)}||_2^2 + ||x_t - x_0^{(j)}||_2^2}{4\sigma_t^2})\mathrm{d}x_t$$

*We will extract the numerator term from the exponential:*

$$||x_t - x_0^{(i)}||_2^2 + ||x_t - x_0^{(j)}||_2^2 = (x_t - x_0^{(i)})^T(x_t - x_0^{(i)}) + (x_t - x_0^{(j)})^T(x_t - x_0^{(j)})$$

$$= 2x_t x_t^T - 2x_t^T(x_0^{(i)} + x_0^{(j)}) + x_0^{(i)T} x_0^{(i)} + x_0^{(j)T} x_0^{(j)}$$

$$= 2||x_t - \frac{x_0^{(i)} + x_0^{(j)}}{2}||_2^2 + x_0^{(i)T} x_0^{(i)} + x_0^{(j)T} x_0^{(j)} - \frac{(x_0^{(i)} + x_0^{(j)})^T (x_0^{(i)} + x_0^{(j)})}{2}$$

*Since the first term related to $x_t$ could be integral to a constant $K$ which do not include $x_0^{(i)}$ and $x_0^{(j)}$.*

$$\int \exp(-\frac{||x_t - \frac{x_0^{(i)} + x_0^{(j)}}{2}||_2^2}{2\sigma_t^2})\mathrm{d}x_t = K$$

*owing to the Gaussian distribution normalization. Then the left term:*

$$x_0^{(i)T} x_0^{(i)} + x_0^{(j)T} x_0^{(j)} - \frac{(x_0^{(i)} + x_0^{(j)T}) (x_0^{(i)} + x_0^{(j)})}{2} = \frac{1}{2}(x_0^{(i)T} x_0^{(i)} + x_0^{(j)T} x_0^{(j)} - 2 \, x_0^{(i)T} x_0^{(j)})$$

$$= \frac{1}{2}||x_0^{(i)} - x_0^{(j)}||_2^2$$

*Substitute in the initial formula, we obtain:*

$$\mathbb{E}_{q(x_t|x_0^{(i)}, y_0^{(i)})} \frac{q(x_t|x_0^{(j)}, y_0^{(j)})}{\sum_k q(x_t|x_0^{(j)}, y_0^{(j)})} \leq \frac{B}{2}(q(y_0^{(i)})q(y_0^{(j)}))^\beta \int \exp(-\frac{||x_t - x_0^{(i)}||_2^2 + ||x_t - x_0^{(j)}||_2^2}{4\sigma_t^2})\mathrm{d}x_t$$

$$= \frac{B}{2}(q(y_0^{(i)})q(y_0^{(j)}))^\beta \int \exp(-\frac{||x_t - \frac{x_0^{(i)} + x_0^{(j)}}{2}||_2^2}{2\sigma_t^2})\mathrm{d}x_t \times \exp(-\frac{||x_0^{(i)} - x_0^{(j)}||_2^2}{8\sigma_t^2})$$

$$= \frac{BK}{2}(q(y_0^{(i)}) \, q(y_0^{(j)}))^\beta \exp(-\frac{||x_0^{(i)} - x_0^{(j)}||_2^2}{8\sigma_t^2})$$

*The K is absorbed in constant B, we finish the proof.*

## D    THE RELATION WITH CBDM LOSS AND ANALYSIS

**Proposition D.1** *When the diffusion model converges, the optimal minimizer $\epsilon^*(x_t, y)$ for the CBDM loss:*

$$\mathcal{L}_{CBDM}(x_t, y, t, \epsilon) = ||\epsilon_\theta(x_t, y) - \epsilon||^2 + \frac{\tau t}{|\mathcal{Y}|}\sum_{y' \in \mathcal{Y}}||\epsilon_\theta(x_t, y) - \epsilon_\theta(x_t, y')||^2$$

*could be denoted as:*

$$\epsilon^*(x_t, y) = \frac{1}{1 + t\tau}\epsilon + \frac{t \, \tau}{(1 + t \, \tau) |\mathcal{Y}|}\sum_{y'} \epsilon_\theta(x_t, y')$$

**Proof D.1** *When the diffusion model converges, for specific label y, the score $\epsilon_\theta(x_t, y')$ for other label $y'$ is fixed. When the loss is minimized:*

$$\mathcal{L}_{CBDM}(x_t, y, t, \epsilon) = ||\epsilon_\theta(x_t, y) - \epsilon||^2 + \frac{\tau t}{|\mathcal{Y}|} \sum_{y' \in \mathcal{Y}} ||\epsilon_\theta(x_t, y) - \epsilon_\theta(x_t, y')||^2$$

$$= \epsilon_\theta(x_t, y)^T \epsilon_\theta(x_t, y) + \epsilon^T \epsilon - 2\epsilon_\theta(x_t, y)^T \epsilon +$$

$$\frac{\tau t}{|\mathcal{Y}|} \sum_{y' \in \mathcal{Y}} (\epsilon_\theta(x_t, y)^T \epsilon_\theta(x_t, y) + \epsilon_\theta(x_t, y')^T \epsilon_\theta(x_t, y') - 2\epsilon_\theta(x_t, y)^T \epsilon_\theta(x_t, y'))$$

$$= (1 + \tau t)\epsilon_\theta(x_t, y)^T \epsilon_\theta(x_t, y) - 2\epsilon_\theta(x_t, y)^T (\epsilon + \frac{\tau t}{|\mathcal{Y}|} \sum_{y' \in \mathcal{Y}} \epsilon_\theta(x_t, y')) + Const$$

*where the Const has no relation with $\epsilon_\theta(x_t, y)$. The term related with $\epsilon_\theta(x_t, y)$ could be expressed as a quadratic form:*

$$\mathcal{L}_{CBDM}(x_t, y, t, \epsilon) = (1 + \tau t)\epsilon_\theta(x_t, y)^T \epsilon_\theta(x_t, y) - 2\epsilon_\theta(x_t, y)^T (\epsilon + \frac{\tau t}{|\mathcal{Y}|} \sum_{y' \in \mathcal{Y}} \epsilon_\theta(x_t, y'))$$

$$= (1 + \tau t)[\epsilon_\theta(x_t, y)^T \epsilon_\theta(x_t, y) - 2 \epsilon_\theta(x_t, y)^T \frac{\epsilon + \frac{\tau t}{|\mathcal{Y}|} \sum_{y' \in \mathcal{Y}} \epsilon_\theta(x_t, y')}{(1 + \tau t)}]$$

$$= (1 + \tau t)[||\epsilon_\theta(x_t, y) - \frac{\epsilon + \frac{\tau t}{|\mathcal{Y}|} \sum_{y' \in \mathcal{Y}} \epsilon_\theta(x_t, y')}{(1 + \tau t)}||_2^2] + const'$$

*When the $\mathcal{L}_{CBDM}$ converges to the minimum, the $L_2$ norm should approximately near zero, so that we have:*

$$\epsilon^*(x_t, y) \approx \frac{1}{1 + t\tau}\epsilon + \frac{t\,\tau}{(1 + t\,\tau)\,|\mathcal{Y}|} \sum_{y'} \epsilon_\theta(x_t, y')$$

**Analysis** This approach has two limitations. Firstly, referring to scores from other labels requires training the entire model within a conditional generation framework, thus restricting its applicability. Secondly, relying on scores from other labels for the same input $(x_t)$ introduces potential biases, particularly when there is a substantial semantic difference between the two classes. This can lead to some degree of offset. In contrast, our method addresses these limitations. First, it can be used in both conditional and unconditional generation scenarios. Second, we utilize the inherent similarity of data in the $L_2$ space where the diffusion model operates. For example, in Figure. 2, we consider the similarity between the red airplane and the red car, ensuring that our enhancement is more logically grounded.

# E   CLASS-WISE FID SCORES OF T2H COMPARISON TO BASE DDPM.

We calculate the FID score using 5k images of T2H and base DDPM for each class on cifar10LT datasets. The final column of the table is overall performance utilizing 50k images.

Table 4: Class-wise FID scores of T2H comparison to base DDPM.

| Class | Airpl | Auto | Bird | Cat | Deer | Dog | Frog | Horse | Ship | Truck | All |
|---|---|---|---|---|---|---|---|---|---|---|---|
| $P_{cls}$ | 0.403 | 0.241 | 0.145 | 0.086 | 0.052 | 0.031 | 0.018 | 0.01 | 0.0067 | 0.004 | 1.0 |
| Base | 31.49 | 15.20 | 40.81 | 32.32 | 32.32 | 36.34 | 40.47 | 23.20 | 26.31 | 23.31 | 10.72 |
| T2H | 31.84 | 14.58 | 19.42 | 28.32 | 18.31 | 26.83 | 29.90 | 17.70 | 21.04 | 22.82 | 6.89(-3.83) |

We have also modified the asssignment of head and tail classes based on whether they are animals or vehicles. As shown in the table below, By altering the assignment of classes, our method demonstrates a greater improvement compared to the baseline.

Table 5: Class-wise FID scores of T2H comparison to base DDPM with shifted categories.

| Class | Horse | Bird | Frog | Deer | Dog | Cat | Truck | Airpl | Auto | Ship | All |
|-------|-------|------|------|------|-----|-----|-------|-------|------|------|-----|
| $P_{cls}$ | 0.403 | 0.241 | 0.145 | 0.086 | 0.052 | 0.031 | 0.018 | 0.01 | 0.0067 | 0.004 | 1.0 |
| Base | 20.19 | 21.99 | 22.45 | 20.57 | 38.07 | 32.73 | 24.87 | 49.24 | 29.48 | 39.12 | 11.79 |
| T2H | 20.16 | 18.52 | 19.83 | 16.23 | 24.60 | 27.50 | 14.73 | 30.08 | 20.99 | 21.41 | 7.15(-4.64) |

## F  THE IMPACT OF DATASET IMBALANCE ON PERFORMANCE.

We have investigated the impact of dataset balance on performance. As illustrated in the following table, we computed the FID scores using different reference sets: one balanced and the other long-tailed sampled. It is observable that, despite the long-tail dataset consisting of real images, its performance metrics are inferior to those of the balanced generated dataset, due to its significant imbalance.

Table 6: The FID (IS) scores for imbalance real dataset and generation dataset. Bal means balanced dataset or generation, ref means reference real dataset

| Bal/no Bal | | No Bal | | | Bal | |
|-----------|-----------|-----|------|-----|-----|------|
| Method | No bal ref | T2H | Base | | T2H | Base |
| Bal ref | 26.22(5.80) | 45.79(7.04) | 46.66(7.03) | | 8.38(9.62) | 12.95(9.52) |
| No-Bal ref | - | 28.68 | 30.34 | | 33.19 | 38.81 |

## G  THE VALIDATION OF TRANSFERRING TARGETS WITH T2H AND H2T.

In this section, we aim to verify the reliability and correctness of the transfer target. Suppose we have a clean sample $x_0$, perturbed with noise $\epsilon \sim \mathcal{N}(0, \sigma_t^2)$ obtaining the noisy sample $x_t$ with probability $p_t = B \ \exp[-\frac{||x_0 - x_t||_2^2}{2\sigma_t^2}]$ where $B \propto \frac{1}{\sigma_t}$ is some normalizing constant.

Then $x_t$ is involving the T2H or H2T algorithm and transfer the target as $x_0^{(z)}$ with probability $p_{sel}(z) = \frac{q(x_t|x_0^{(z)}, y_0^{(z)})}{\sum_j q(x_t|x_0^{(j)}, y_0^{(j)})}$ where $q(x_t|x_0^{(z)}, y_0^{(z)}) = B \ \exp[-\frac{||x_0^{(z)} - x_t||_2^2}{2\sigma_t^2}]$ also calculated with Gaussian kernel. Firstly, let us discuss the similarity of $x_0$ and $x_0^{(z)}$ measured with $||x_0 - x_0^{(z)}||_2^2$ with following proposition.

**Proposition G.1** *The L2 similarity with $x_0$ and $x_0^{(z)}$ bounded by the $p_t$ and $p_{sel}(z)$:*

$$||x_0 - x_0^{(z)}||_2^2 \leq -2\sigma_t^2(\log(p_t \ p_{sel}(z)) + \log(p_t + p_{sel}(z)) + 2\log\sigma_t). \qquad (18)$$

**Proof G.1**

$$||x_0 - x_0^{(z)}||_2^2 \overset{1}{\leq} -2\sigma_t^2(-\frac{||x_0^{(z)} - x_t||_2^2}{2\sigma_t^2} - \frac{||x_0 - x_t||_2^2}{2\sigma_t^2})$$

$$= -2\sigma_t^2(\log\exp[-\frac{||x_0^{(z)} - x_t||_2^2}{2\sigma_t^2}] + \log\exp[-\frac{||x_0 - x_t||_2^2}{2\sigma_t^2}])$$

$$= -2\sigma_t^2(\log\frac{p_t \ p_{sel}(z) \ \sum_j q(x_t|x_0^{(j)}, y_0^{(j)})}{B^2})$$

$$\overset{2}{\leq} -2\sigma_t^2(\log(p_t \ p_{sel}(z)) + \log(p_t + p_{sel}(z)) + 2\log\sigma_t)$$

*where 1 involves utilizing the triangle inequality. And 2 comes from $\sum_j q(x_t|x_0^{(j)}, y_0^{(j)}) \geq q(x_t|x_0, y_0) + q(x_t|x_0^{(z)}, y_0^{(z)})$.*

As can be observed from the proposition above, the similarity $x_0$ between and $x_0^{(z)}$ is bounded by an upper limit. Moreover, in the sampling process, the larger the values $p_t$ and $p_{sel}(z)$ the tighter this bound becomes.

Furthermore, from an alternative viewpoint, we evaluate the probability of obtaining $x_t$ using a clean $x_0^{(z)}$ within the framework of a one-to-one clean-to-noisy mapping.

**Proposition G.2** *The probability $p_t^z$ of obtaining $x_t$ from $x_0^{(z)}$ under single denoising target scenario could be:*

$$p_t^z \geq \frac{p_{sel}(z)\, p_t}{1 - p_{sel}(z)}, \tag{19}$$

*where $p_{sel}(z)$ comes from Equation.( 8) and $q(x_t|x_0, y_0) \propto B \exp[-\frac{||x_0^{(z)} - x_t||_2^2}{2\sigma_t^2}]$*

**Proof G.2**

$$p_{sel}(z) = \frac{q(x_t|x_0^{(z)}, y_0^{(z)})}{\sum_j q(x_t|x_0^{(j)}, y_0^{(j)})}$$

$$\leq \frac{q(x_t|x_0^{(z)}, y_0^{(z)})}{q(x_t|x_0, y_0) + q(x_t|x_0^{(j)}, y_0^{(j)})},$$

*because $p_t = q(x_t|x_0, y_0)$ and $p_t = q(x_t|x_0^{(z)}, y_0^{(z)})$ under the assumption of Equation.( 8) so that:*

$$p_{sel}(z) \leq \frac{p_t^z}{p_t^z + p_t}$$

*so we obtain:*

$$p_t^z \geq \frac{p_{sel}(z)\, p_t}{1 - p_{sel}(z)}.$$

We have obtained a lower bound for $p_t^z$. If we assume $p_t \approx p_{sel}(z) \approx 0.5$ then the $p_t^z \geq 0.5$ where is also valid noise-clean pair for $x_t \to x_0^{(z)}$ under the single target scenario.

## H  EXPERIMENTS ON IMAGENETLT DATASET

We conduct our method on large scale datasets ImagenetLT, the performance is shown in the following Table. 7. We generate 20k images with 1000 classes and use the balanced validation set with 20k images as the reference set for the calculation of FID scores. As shown in the Table, Our results on large-scale are consistent to the observations on small-scale data, which validates the effectiveness of our method at scale.

Table 7: Experiments on large scale dataset on ImagenetLT

| Method | FID | IS |
|---|---|---|
| base DDPM | 26.95 | **15.99** |
| CBDM | 28.12 | 15.86 |
| T2H (Ours) | **25.42** | 15.96 |

## I  FINETUNING EXPERIMENTS FROM NORMAL TRAINED MODEL

We also conduct the experiments of finetuning with the model pre-trained with normal denoising loss function. The finetuning starting step is ranging from 100k to 500k, and the results with no pretriaining and no finetuning are also provided. The conditional model is finetuned with T2H strategy, while the unconditional model is finetuned with H2T strategy.

As shown in the table, finetuning is capable of further improving model performance based on pre-training. However, as the number of pretraining steps increases, the extent of improvement gradually diminishes and eventually becomes stable.

Table 8: Finetuning from normal pretrained models with different training steps.

| Pretrained | Pretrained Steps | No finetune | No pre-train | 100k | 200k | 300k | 400k |
|---|---|---|---|---|---|---|---|
| Base Uncond | FID | 25.31 | 16.09 | 18.13 | 20.31 | 21.65 | 21.10 |
| | IS | 7.01 | 8.27 | 7.93 | 7.86 | 7.32 | 7.33 |
| Base Cond | FID | 10.20 | 6.89 | 7.48 | 7.87 | 8.01 | 8.12 |
| | IS | 9.25 | 9.75 | 9.64 | 9.63 | 9.62 | 9.56 |

## J  IMPLEMENTATION DETAIL OF TOY GAUSSIAN EXAMPLES IN FIGURE. 4

We random sample 10k samples from distribution $\mathcal{N}((0, 4), 0.2I)$ as head samples $\{x_0^{(H_i)}\}$, and 0.1k samples from $\mathcal{N}((0, -4), 0.2I)$ simulating tail samples $\{x_0^{(T_j)}\}$. The empirical distribution of the overall dataset could be denoted as:

$$p_{data}(x) = \sum_i \delta(x - x_0^{(H_i)}) + \sum_j \delta(x - x_0^{(T_j)})$$

Simulating the forward diffusion process, we convolute the empirical distribution with Gaussian noise $\mathcal{N}(0, \sigma_t^2 I)$:

$$p_{\sigma_t}(x_t) = p_{data}(x) * \mathcal{N}(0, \sigma_t^2 I) = \sum_i \mathcal{N}(x_t; x_0^{(H_i)}, \sigma_t^2 I) + \sum_j \mathcal{N}(x_t; x_0^{(T_j)}, \sigma_t^2 I) \tag{20}$$

The score estimation is calculated as $\nabla_{x_t} \log p_{\sigma_t}(x_t)$, we substitute the result in Eq. (20):

$$\nabla_{x_t} \log p_{\sigma_t}(x_t) = \nabla_{x_t} \log(\sum_i \mathcal{N}(x_t; x_0^{(H_i)}, \sigma_t^2 I) + \sum_j \mathcal{N}(x_t; x_0^{(T_j)}, \sigma_t^2 I))$$

$$= \frac{1}{\sum_i \mathcal{N}(x_t; x_0^{(H_i)}, \sigma_t^2 I) + \sum_j \mathcal{N}(x_t; x_0^{(T_j)}, \sigma_t^2 I)} \times$$
$$\nabla_{x_t}(\sum_i \mathcal{N}(x_t; x_0^{(H_i)}, \sigma_t^2 I) + \sum_j \mathcal{N}(x_t; x_0^{(T_j)}, \sigma_t^2 I))$$

$$= -\frac{1}{\sigma_t^2} \frac{\sum_i (x_t - x_0^{(H_i)}) \mathcal{N}(x_t; x_0^{(H_i)}, \sigma_t^2 I) + \sum_j (x_t - x_0^{(T_j)}) \mathcal{N}(x_t; x_0^{(T_j)}, \sigma_t^2 I)}{\sum_i \mathcal{N}(x_t; x_0^{(H_i)}, \sigma_t^2 I) + \sum_j \mathcal{N}(x_t; x_0^{(T_j)}, \sigma_t^2 I)}$$

The last step is because of:

$$\nabla_{x_t} \mathcal{N}(x_t; x, \sigma_t^2 I) = \nabla_{x_t} C \exp(-\frac{||x_t - x||_2^2}{2\sigma_t^2})$$

$$= -C \exp(-\frac{||x_t - x||_2^2}{2\sigma_t^2}) \nabla_{x_t} \frac{||x_t - x||_2^2}{2\sigma_t^2}$$

$$= -\frac{1}{\sigma_t^2} \mathcal{N}(x_t; x, \sigma_t^2 I)(x_t - x)$$

The number of points in the head class is 100 times greater than the tail class, resulting in an imbalance factor of 0.01. It can be observed that the score distribution across the entire space is dominated by the head class, leading to a highly imbalanced dataset generation.

## K  FUTURE WORK

Now, diffusion models are widely applied to the generation and learning research in multimodal fields. Because the definition of the long-tail problem is not yet clear in the context of multimodal data, more efforts can be used to define the data defects in multimodal data (Chen et al., 2023a) such as long tail distribution, how to solve the long-tail generation problem in multimodal scenarios, and more downstream tasks in multimodal contexts (Zhang et al., 2024).

## L  VISUALIZATION RESULTS

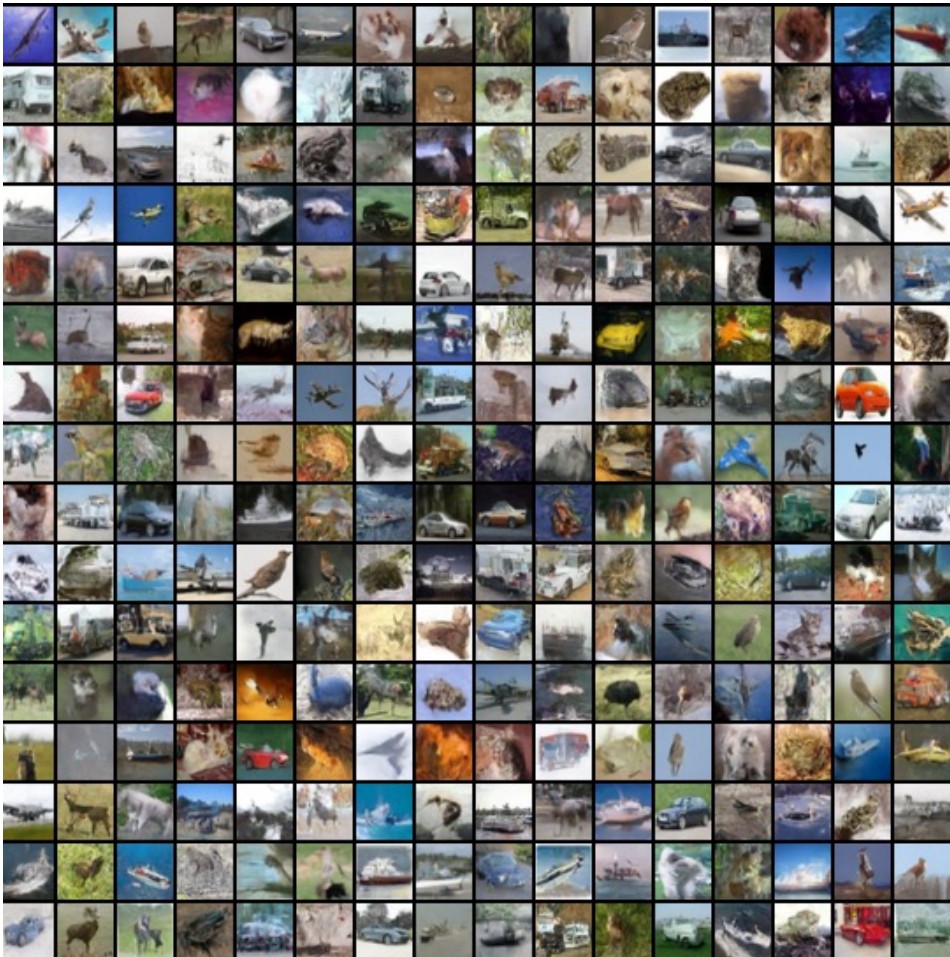

Figure 8: Visualization of generation results of CIFAR10LT dataset with T2H conditional generation

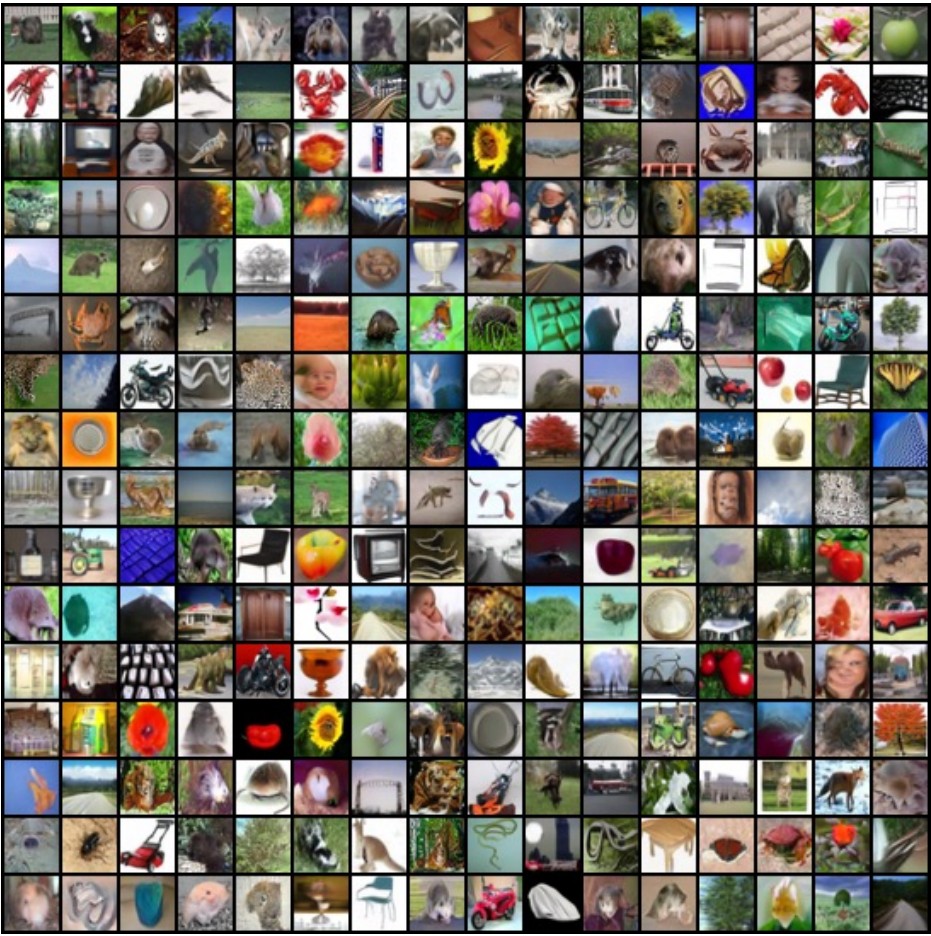

Figure 9: Visualization of generation results of CIFAR100LT dataset with T2H conditional generation

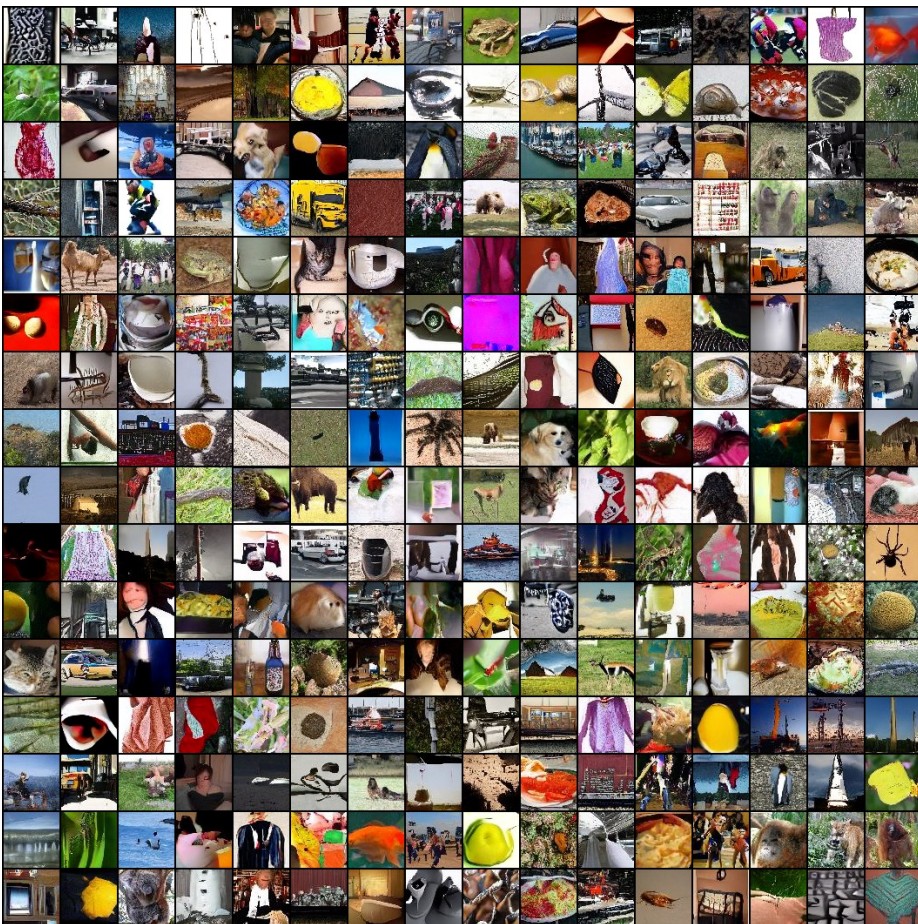

Figure 10: Visualization of generation results of TinyImageNet200 dataset with T2H conditional generation

