# OpenReview forum: "Long-tailed Diffusion Models with Oriented Calibration"
_ICLR.cc/2024/Conference — ICLR 2024 poster_

### Official Review · Reviewer_Uefc · 2023-10-27

**Soundness:** 4 excellent
**Presentation:** 4 excellent
**Contribution:** 4 excellent
**Rating:** 6
**Confidence:** 4

**Summary:**

In this paper, the author's primary focus is on tackling the diffusion model generation tasks in the context of long-tailed training scenarios. Different from previous works that enhance tail generation by relying on the abundant diversity derived from the head class (the condition capacity of the model prediction), this paper directly establishes the knowledge transfer from head data samples, based on the multi-objective characteristics of the score function in the diffusion process. A directional calibration for the estimation of noisy tail sample score is performed towards the clean head samples (T2H), leveraging the similarity within the data distribution from head to tail classes. Meanwhile, H2T is proposed for the unconditional generation. Extensive experiments demonstrate the effectiveness of the proposed method.

**Strengths:**

1. This paper presents a novel idea that calibrates the estimation of noisy tail sample score by the clean head samples instead of relying on the condition capacity of the model prediction.
2. This paper has a well-organized structure which makes it easy for readers to understand the research.
3. The paper is well-supported by a strong theoretical foundation, which robustly underpins the proposed solutions.
4. The text is clear with a good writing style.

**Weaknesses:**

1. The explanations of some figures are not sufficiently comprehensive. For instance, the author could enhance the clarity of differentiation by adding annotations or captions to both the top and bottom parts of Figure 1.
2. Some problems, which I will raise in the following section.

**Questions:**

1. In the paper, T2H is utilized to estimate noisy tail samples using clean head samples, and this process is explained as a way to enhance the performance of long-tailed conditional generation. However, I have difficulty understanding why H2T, which estimates noisy head samples using clean tail samples, would improve the performance of unconditional generation. This is because, in my view, the style of the tail samples should be relatively homogeneous and might not provide substantial knowledge for estimating the head samples. Could the authors please provide an explanation?
2. In Table1, why the "Full" (allowing both transfer directions) performs worse than the H2T and T2H (only allowing one direction) under the unconditional generation? Meanwhile, I'd like to see the performance of "Full" for the conditional generation.
3. The author should explain more about why the baseline DDPM has a higher IS than T2H (Table 2 and Table 3).

---

> ### Author Response · Authors · 2023-11-18
> **Response to Reviewer Uefc**
>
> ### W1
> > The explanations of some figures are not sufficiently comprehensive. For instance, the author could enhance the clarity of differentiation by adding annotations or captions to both the top and bottom parts of Figure 1
>
> Thank you for the suggestion. We follow the reviewer's advice and have improved the caption of the figures including Figure 1 in the revised submission.
>
> ### Q1
> > In the paper, T2H is utilized to estimate noisy tail samples using clean head samples, and this process is explained as a way to enhance the performance of long-tailed conditional generation. However, I have difficulty understanding why H2T, which estimates noisy head samples using clean tail samples, would improve the performance of unconditional generation. This is because, in my view, the style of the tail samples should be relatively homogeneous and might not provide substantial knowledge for estimating the head samples. Could the authors please provide an explanation?
>
> Thanks for the comments. In unconditional generation, the most difficult challenge to tackle is **the balanced generation**. Unlike the conditional generation that could be achieved by sampling labels with balance, as shown in the table below, on the imbalanced cifar10lt dataset, the FID score with a balanced reference set (26.22) is significantly higher compared to that based on conditional generation (8.38).  However, training an unconditional model directly with a long tail dataset means that in the space, **the score for a noisy sample is predominantly influenced by the head samples**. The H2T strategy enhances the tendency to generate tail samples, thereby better ensuring the balance of the generated dataset.
>
>
> **FID scores&thinsp;$\downarrow$ with balanced and non-balanced datasets**
>
> |    Reference Dataset     | No-Bal reference Set | Bal Generation DataSet |
> |:------------------------:|:--------------------:|:----------------------:|
> | No-Bal reference DataSet |          -           |         33.19          |
> |  Bal reference DataSet   |        26.22         |          8.38          |
>
> More experimental details could be found in Appendix. G.
> ### Q2
> > In Table1, why the "Full" (allowing both transfer directions) performs worse than the H2T and T2H (only allowing one direction) under the unconditional generation? Meanwhile, I'd like to see the performance of "Full" for the conditional generation.
>
> Thank you for the comments. The answers are listed below:
> - **'Full' worse then T2H and H2T**
> For unconditional generation, as former discussed, H2T could alleviate the un-balanced problem and T2H could improve the head-to-tail knowledge transfer.
> However, the 'Full' mode is not a mere overlay of H2T and T2H strategies. This is because, during the score mixing as described in Equation 6, it is impossible to simultaneously increase the contributions of both head class and tail class samples under the condition of Eq. 7. The integration of these two methods is a challenge we plan to address in our future work.
>
> - **'Full' mode in condition generation**  We have added the corresponding results in Table 1 (i.e. model I). According to the results, we can see that the performance of 'Full' is better than H2T but weaker than T2H under conditional generation. The T2H still remains the most effective method.
> ### Q3
> > The author should explain more about why the baseline DDPM has a higher IS than T2H (Table 2 and Table 3).
>
> Thanks for the comments. This is a really interesting phenomenon and need to find clues within the calculation method of IS. The Inception scores:
> $$
> IS(G)=\exp \mathbb{E} _{x \sim g} D _{KL}(p(y|x)||p(y)))
> \approx \exp \frac{1}{N}\sum _i D _{KL}(p(y|x _i)||p(y))
> $$
> The each term in exponential is a KL divergence between the label distribution $p(y|x_i)$ for each sample and the overall balanced distribution. When the $p(y|x_i)$ is sharper and more concentrates to one class, the KL divergence become larger.
> Since both our method and baseline method CBDM involve the score transfer among classes, the probability distribution $p(y∣x)$ might not be as sharp as in the base DDPM, especially in scenarios involving a large number of categories. This account for a slight decrease in performance of IS score, which is also observed in larger dataset ImageNetLT:
>
> | Method    | FID &thinsp;$\downarrow$  | IS &thinsp;$\uparrow$    |
> |:---------:|:-----:|:------:|
> | base DDPM | 26.95 | **15.99**  |
> | CBDM      | 26.12 | 15.86  |
> | T2H(Ours) | **25.42** | 15.96  |

---

### Official Review · Reviewer_PNyr · 2023-10-30

**Soundness:** 2 fair
**Presentation:** 1 poor
**Contribution:** 2 fair
**Rating:** 6
**Confidence:** 4

**Summary:**

This paper presents an algorithm for calibrating diffusion models for long-tailed data. Based on the recent work by Xu et al., 2023, in this work, the target for the score estimation is re-written as a mixture of multiple targets computed from a reference batch. The reference batch, unlike the work of Xu et al,. 2023, is sampled from a class-balanced distribution in order to calibrate the model for tail-class samples. For the conditional generation, the authors further propose a strategy called Tail-to-Head (T2H), where the target of the score estimation for a tail class sample is chosen to be a clean head class sample. The paper also suggests a variant called Head-to-Tail (H2T), where in this case a noisy head sample takes a clean tail sample as a target. The proposed method is demonstrated to improve image fidelity when applied to popular imbalanced image benchmarks.

**Strengths:**

- The paper is tackling an interesting problem.
- The proposed method is simple and easy to implement.
- The experimental results, at least for the simple benchmarks, is promising.

**Weaknesses:**

- The presentation is poor. I don't really follow the description of the intuition behind the presented method. Please see my questions below.
- Again, probably related to the first point, I don't see how the proposed method yields a valid regression target. The algorithm seems to be a heuristics without considering correctness.
- Batch-Resample cannot be considered a novel contribution, as it is one of the standard approaches one could take for a problem involving a long-tailed dataset (e.g., long-tailed image classification).

**Questions:**

- I'm very confused about the argument around Proposition 3.1. First, it is not clear from the text what the "score weight" or "weight of score" means and why it is important. I guess this term refers to the mixing coefficients (normalized transition probabilities) appearing in the multi-target representation of the score function. Secondly, what is the author trying to make out of Proposition 3.1? How is it related to the algorithm? The sentence "Intuinitively (intuitively, I guess), improving the score weight of head-class samples for the noisy $x_t$ from tail-class samples, increases the generation diversity of tail categories" is not clear at all. What is "improving" the score weight? Is it increasing the value of the score weight? If so, how is it related to increasing the generation diversity?
- The introduction of the selection procedure (equation (11)) comes without sufficient explanation. If I have to understand it, the random sampling for $z$ can be understood as a Monte-Carlo estimator of equation (8); But then the proposed algorithm alters the target score if $q(z_y) \geq q(y_i)$; as a result, the target of the score estimation becomes something different, yielding a biased estimator of the original target. Is this intended? If so, how do you guarantee that the proposed algorithm is learning a correct target?
- In the beginning you assumed $q(x_t|x_0, y_0) \propto B q(y_0)^\beta \exp(-\Vert x_t-x_0\Vert_2^2/2\sigma_t^2)$, but $p_\text{sel}(z)$ described to computed only with $C \exp(-\Vert x_z-x_t\Vert_2^2/2\sigma_t^2)$ without $q(y)^\beta$ term. Is this a typo or am I missing something here?
- Similarly, I don't get the intuition behind the H2T, since from the beginning I failed to follow how Proposition 3.1 is linked to the T2H. What is the meaning of setting $\beta=-1$ and why is it beneficial for unconditional sampling?
- The experiments mostly show the results of T2H on class conditinal generation and H2T for unconditional generation, but how do they work for opposite settings (T2H for unconditional and H2T for conditional)? Is the direction of transfer matters? Or both direction helps in either cases, but the effectiveness of transfer may vary depending on the task?

---

> ### Author Response · Authors · 2023-11-18
> **Response to Reviewer PNyr[1/2]**
>
> ### Q1 & W1
> > The presentation is poor. I don't really follow the description of the intuition behind the presented method. Please see my questions below.
>
> > I'm very confused about the argument around Proposition 3.1. First, it is not clear from the text what the "score weight" or "weight of score" means and why it is important. I guess this term refers to the mixing coefficients (normalized transition probabilities) appearing in the multi-target representation of the score function. Secondly, what is the author trying to make out of Proposition 3.1? How is it related to the algorithm?
>
> Thanks for the comment.
> - **About score weight**
> 1. We define the term "score weight" or "weight of score" in Equation (6).
> 2. Importance of the score
>     i) From the perspective of methodology, this weight is derived from the estimation of the score function used in the denoising model.
>     ii) We use Proposition 3.1 to show the potential challenges in allowing diffusion models trained on such a long-tailed distribution with adjusting this weight to produce sample distributions with balanced labels.
>     iii) Empirically, our experiments in Table 2 also demonstrate the performance can be significanly improved using the score weight in this context.
>
>
> - **About Proposition3.1**
> We agree the Proposition 3.1 is not suitable for introducing the T2H method. "In fact, the Proposition **sets an upper limit on the transfer strength between different categories**, particularly for head-to-tail class transfer. Actually, this proposition is more suitable for highlighting the importance of batch resampling. That is to say, if we increase the sampling probability of the head class based on Eq. 7, then without batch resampling, the strength of the head-to-tail transfer would be constrained.
>
>
>
>
> > The sentence "Intuinitively (intuitively, I guess), improving the score weight of head-class samples for the noisy $x_t$ from tail-class samples, increases the generation diversity of tail categories" is not clear at all. What is "improving" the score weight? Is it increasing the value of the score weight? If so, how is it related to increasing the generation diversity?
>
> - We re-write and re-organize the section here, and trying to make the illustration more clear. In essence, our objective here is to enable the score of the current noisy tail sample, as demonstrated in Proposition 3.1,  encourging the head-to-tail transfer in order to promote the tail generation diversity.
>
>
> ### W2 and Q2
> >Again, probably related to the first point, I don't see how the proposed method yields a valid regression target. The algorithm seems to be a heuristics without considering correctness.
>
> > The introduction of the selection procedure (equation (11)) comes without sufficient explanation. If I have to understand it, the random sampling for $z$ can be understood as a Monte-Carlo estimator of equation (8); But then the proposed algorithm alters the target score if $q(z_y) \geq q(y_i)$; as a result, the target of the score estimation becomes something different, yielding a biased estimator of the original target. Is this intended? If so, how do you guarantee that the proposed algorithm is learning a correct target?
>
> Thanks for the comments. In fact, theoretically we want to acquire a $\nabla_{x_t}\log q^{\star}(x_t|y)$ under full balanced distribution so it is truely a biased estimator of initial $\nabla_{x_t}\log q(x_t|y)$ with long tailed distribution. Empirically, we improve the head class samples contribution in Eq. 6 to augment the tail sample generation. We first transform the score mixing as a score selection way and then alter the target intentionally. We verify the validity and correctness of our target transfer from three aspects:
> - The score transfer here is based on a Gaussian Kernel $\exp - \frac{||x_t-x_0||^2_2}{2\sigma_t^2}$, which ensures a certain degree of **similarity between the transfer target and the original image** for a given time step. As detailed discussed in the Appendix H.1, **the L2 norm of transferred target and initial target** is constrained by an upper limit of sampling probability:
> $$||x _0-x _z|| _2^2  \leq -2\sigma _t^2(\log (p _t ~ p _{sel}(z))+\log (p _t+p _{sel}(z)) + 2\log \sigma _t)$$
> here $p _t$ and $p _{sel}(z)$ are the two sampling probability during the training process.
> - Furthermore, under the single target denoising condition, the current noisy sample **$x_t$ still has a relatively high probability $p_t^z$ of being obtained by adding noise to the transferred target**, as detailed illustrated in Appendix H.2:
> $$p_t^z \geq \frac{p_{sel}(z)~p_t}{1-p_{sel}(z)}.$$
> The equation means the current $x_t$ is also likely to obatain from $x_z$ under the single one-to-one noisy clean pairs training strategy.
> - Our target transfer only occurs after 400 steps; essentially, there is no transfer within the first 400 steps, as shown in the left subfigure of Figure. 6. This evidence also make sure the stability of the target.

---

> ### Author Response · Authors · 2023-11-18
> **Response to Reviewer PNyr[2/2]**
>
> ### W3
> > Batch-Resample cannot be considered a novel contribution, as it is one of the standard approaches one could take for a problem involving a long-tailed dataset (e.g., long-tailed image classification).
>
> We recognize that Batch Re-sample is a widely adopted technique for addressing long-tail issues. However, its application and impact in the context of long-tailed generative modeling, especially within diffusion models, remain underexplored. Nonetheless, the application of Batch Re-Sample in long-tail generation remains an un-explored area, especially **concerning its potential to alleviate the score dominance of head class samples in unconditional generation and to facilitate the transfer from head to tail classes** in conditional generation, which is studied firstly in our work.
>
> ### Q3
> > In the beginning you assumed $q(x_t|x_0, y_0) \propto B q(y_0)^\beta \exp(-\Vert x_t-x_0\Vert_2^2/2\sigma_t^2)$, but $p_\text{sel}(z)$ described to computed only with $C \exp(-\Vert x_z-x_t\Vert_2^2/2\sigma_t^2)$ without $q(y)^\beta$ term. Is this a typo or am I missing something here?
>
> Sorry for the confusion. Our aim is to reasonably increase the contribution of head samples in score mixing. Therefore, after transforming score mixing into score selection, we adopted a more direct method. **The approach of T2H permits targets to transfer only to the head classes, but exclusively under the L2 distance of the Gaussian Kernel**. Naturally, we also conducted experiments based on the Eq. 7 with $q(y_0)$ involved, as shown in the Figure 4. The result of $\beta=1$ aligns perfectly with T2H. In the new version of the section, we are trying to make it more clear.
>
> ### Q4
> > Similarly, I don't get the intuition behind the H2T, since from the beginning I failed to follow how Proposition 3.1 is linked to the T2H. What is the meaning of setting $\beta=-1$ and why is it beneficial for unconditional sampling?
>
> Thank you for the comments. The answers are listed below:
> - **About $\beta=-1$ for H2T**
> When beta is set to -1, the smaller $q(y_T)$of tail class samples results in an increased probability of $q(x_t|x_T,y_T)$ from Eq. 7 in the batch as $x^{-1}$ is a decreasing function. Consequently, for a noisy head sample, **the contribution of the tail sample targets are enhanced**, corresponding to H2T algorithm.
> - **Why $\beta=-1$ is beneficial for unconditional generation**
> In unconditional generation, unlike conditional generation where labels are involved, **achieving balance is the greatest challenge**. As shown in the table below, for the imbalanced cifar10lt dataset we used, the FID score with a balanced reference set is significantly higher compared to a balanced dataset based on conditional generation. Therefore, the primary challenge in unconditional generation is to **generate samples more evenly across all categories**. However, training an unconditional model directly with a long tail dataset means that in the space, the score for a noisy sample is **predominantly influenced by the head samples**. To address this, we incorporate the H2T (Head to Tail) strategy, which enhances the tendency to generate tail samples, thereby better **ensuring the balance of the generated dataset**.
>
> **FID scores&thinsp;$\downarrow$ with balanced and non-balanced datasets**
> |    Reference Dataset     | No-Bal reference Set | Bal Generation DataSet |
> |:------------------------:|:--------------------:|:----------------------:|
> | No-Bal reference DataSet |          -           |         33.19          |
> |  Bal reference DataSet   |        26.22         |          8.38          |
> ### Q5
> > The experiments mostly show the results of T2H on class conditinal generation and H2T for unconditional generation, but how do they work for opposite settings (T2H for unconditional and H2T for conditional)? Is the direction of transfer matters? Or both direction helps in either cases, but the effectiveness of transfer may vary depending on the task?
>
> Thank you for the comments. The answers are listed below:
> - **Conditional and Unconditional work for opposite settings**
> As previously discussed, achieving a balanced generation is crucial in the realm of unconditional generation. The 'Head to Tail' (H2T) approach effectively **balances the generation tendency towards tail classes**. In contrast, for conditional generation, attaining label balance is more straightforward, as it can be managed by generating an equal number of samples for each class. However, the primary challenge in conditional generation lies in the **limited diversity of samples in the tail categories**.
>
> - **Effectiveness of transfer**
> Yes, both direction helps in either cases. However, owing to **the difference for the main challenge of unconditional and conditional tasks**, the effectiveness of the H2T and T2H approaches varies under the two different scenarios

---

> > ### Comment · Reviewer_PNyr · 2023-11-21
> > **Response**
> >
> > I appreciate the authors' response which resolves some of my concerns (especially on presentation);
> > I strongly recommend enhancing the presentation further. I increase my score accordingly.

---

> > > ### Author Response · Authors · 2023-11-21
> > >
> > > We do appreciate the reviewer's positive support and the further advice on presentation. We will seriously take the reviewer's advice to carefully improve our writing and presentation of the whole paper, and promise that the finally updated version will be more clear and readable by incorporating all comments on writing. Thank you very much again.
> > >
> > > Sincerely,
> > > The authors of Submission 1786

---

### Official Review · Reviewer_i52Z · 2023-10-31

**Soundness:** 2 fair
**Presentation:** 3 good
**Contribution:** 2 fair
**Rating:** 6
**Confidence:** 4

**Summary:**

This paper addresses the problem of long tail diffusion model generation. In this paper, a directional calibration for the estimation of
noisy tail sample score is performed towards the clean head samples (T2H), leveraging the similarity within the data distribution from head to tail classes.

**Strengths:**

1. This paper propose a strategy denoted as ”Batch Resample” to sample a more balanced reference batch.
2.This paper has developed a method denoted as T2H based on the multi-target nature of score estimation to effectively calibrate and enhance the generation of tail classes in the semantic formation period, thereby significantly improving the overall generation performance.

**Weaknesses:**

1. The paper introduces two optimization methods, Batch Resample and T2H. The Batch Resample is a common solution to the long-tailed problem, and it does not seem to bring any novelty.
2. There is a lack of connection between the proof and implementation of T2H, especially the transition from equation 10 to equation 11.
3. The author lacks experimental validation on large-scale datasets.

**Questions:**

1. T2H enhances the quality of image generation for the current category by calculating the multi-nominal distribution in Equation 11, and then samples $z$ from this distribution, and selecting categories $y_z$ with a sample density greater than that of the category $y_i$. While the authors provide detailed theoretical proof, the transition from equation 10 to equation 11 seems to lack coherence. When sampling z, why not model the distribution based on the similarity between the original images (the current sample image and other images in the batch, as in equation 10), rather than through the noisy version of the current sample image and other images in the batch?
2. In the T2H method mentioned, the distribution $p_{sel}(z)$ adds weight to the other images within the batch, encouraging similar images to have a greater influence on the current sample. So, what would happen if we do not consider the distribution $p_{sel}(z)$ and assume that all images will have an effect? Also, if we keep the distribution $p_{sel}(z)$ unchanged and assume that all $y_z$ will have an impact, not just those with a high sample density, would this be more reasonable?
3. The authors seem to lack an analysis in their experiments regarding how much growth the proposed solutions have brought to both the head and tail categories in the dataset, respectively.
4. How effective are the proposed methods on larger datasets (ImageNet-LT)?

---

> ### Author Response · Authors · 2023-11-18
> **Response to Reviewer i52Z[1/2]**
>
> ### W1
> > The paper introduces two optimization methods, Batch Resample and T2H. The Batch Resample is a common solution to the long-tailed problem, and it does not seem to bring any novelty.
>
> We recognize that Batch Re-sample is a widely adopted technique for addressing long-tail issues. However, its application and impact in the context of long-tailed generative modeling, especially within diffusion models, remain underexplored. In Proposition 3.1, we introduce an upper bound formulation for the rebalancing process. This framework ensures the feasibility of optimization when batch rebalancing is employed in these models.
>
>
> ## W2 & Q1
> > There is a lack of connection between the proof and implementation of T2H, especially the transition from equation 10 to equation 11.
>
> > T2H enhances the quality of image generation for the current category by calculating the multi-nominal distribution in Equation 11, and then samples $z$ from this distribution, and selecting categories $y_z$ with a sample density greater than of the category $y_i$. While the authors provide detailed theoretical proof, the transition from equation 10 to equation 11 seems to lack coherence. When sampling z, why not model the distribution based on the similarity between the original images (the current sample image and other images in the batch, as in equation 10), rather than through the noisy version of the current sample image and other images in the batch?
>
> Thanks for your comments. The answers are listed below:
> - **Lack of coherence from Eq.10 to Eq.11**
> Thank you for pointing out this gap and we agree with this point. To Illustrate the 'Tail to Head' (T2H) concept more clearly, we have rewritten this section and incorporated illustrative diagrams, as shown in Figure 3. The motivation of T2H is to **improve the head samples contributions in the tail sample generation**. For Proposition 3.1, We believe that it is more appropriate to **demonstrate the enhancement for the strength of the transition from head to tail for batch resample**.
>
> - **Clean sample similarity**
> Utilizing the similarity between head and tail samples is indeed one of the core motivations of our work. However, the diffusion model generates images with a denoising process. Our approach make use of this evidence to different noise levels for better capture the similarity between them. Actually, **the transferred clean sample $x_z$ obtained using our method exhibits a certain degree of similarity with the original clean target $x_0$, measured with L2 norm** as:
> $$||x _0-x _z|| _2^2  \leq -2\sigma _t^2(\log (p _t ~ p _{sel}(z))+\log (p _t+p _{sel}(z)) + 2\log \sigma _t)$$
> where $p _t$ and $p _{sel}(z)$ are the two sampling probability during the training process. The detailed proof is provided in Proposition H.1in Appendix. H.
>
> The sampling distribution is derived from Equations (5) and (6), with Equation (10) from Proposition 3.1 serving to interpret this distribution. That is, the sampling distribution is influenced by both the label $q(y)$ and clean image similarity. To clarify this concept, we made the following adjustments:
>
> - We move Eq (11) to the paragraph ahead of Eq (10). Now it is placed in Eq (8) and we use Figure 3 to illustrate the $p_{sel}(z)$.
>
> - We move proposition 3.1 in front of the paragraph of Batch Re-sample, explaining the motivation of batch resample technique.
>
>
> ### Q2
> > In the T2H method mentioned, the distribution $p_{sel}(z)$ adds weight to the other images within the batch, encouraging similar images to have a greater influence on the current sample. So, what would happen if we do not consider the distribution $p_{sel}(z)$ and assume that all images will have an effect? Also, if we keep the distribution $p _{sel}(z)$ unchanged and assume that all $y_z$ will have an impact, not just those with a high sample density, would this be more reasonable?
>
> Thanks for your comments. Actually, considering the effect from all samples on the present score is precisely what Eq. 6 illustrates. The $p _{sel}(z)$ comes from the normalized weight from the individual score in Eq. 6 and we **transform the weighted mixing problem to a score selection problem with the mixing weight as the selection probability**. I don't quite understand what is "do not consider the distribution $p _{sel}(z)$". Is that means $p _{sel}(z)$ follows a certain distribution?
>     1) if $p _{sel}(z)$ follows a delta distribution $\delta _z^i$, which means the $p _{sel}^z = 1$ when $z=i$, then this would degrade into base DDPM.
>     2) if $p _{sel}(z)$ follows a uniform distribution(is this all $y _z$ have a impact?) without weighted, then the score is estimated with all scores averaging among all samples, then the generation will reach the average of the dataset.
>
> If we assume all $y_z$ has a impact based on Eq. 6 instead of samples with higher densities, that is the $\beta=0$ in Figure 5. The performance is weaker than directly utilizing $\beta=1$ or T2H for conditional generation.

---

> ### Author Response · Authors · 2023-11-18
> **Response to Reviewer i52Z[2/2]**
>
> ### Q3
> > The authors seem to lack an analysis in their experiments regarding how much growth the proposed solutions have brought to both the head and tail categories in the dataset, respectively.
>
> Thank you very much for the advice. We have added a detailed performance table for seperate classes on CIFAR10LT, where the $q(y)$ is the class frequency for discriminating the head and tail categories.
>
> | Class   | Airpl | Auto  | Bird  | Cat   | Deer  | Dog   | Frog  | Horse | Ship   |
> |---------|-------|-------|-------|-------|-------|-------|-------|-------|--------|
> | $q(y)$   | 0.403 | 0.241 | 0.145 | 0.086 | 0.052 | 0.031 | 0.018 | 0.01  | 0.0067 |
> | BaseFID | 31.49 | 15.20  | 40.82  | 32.32 | 32.32 | 36.34 | 40.47 | 23.21  | 26.31  |
> | T2H     | 31.84 | 14.58 | 19.42 | 28.32 | 18.31  | 26.83 | 29.91  | 17.70  | 21.04  |
>
> As you could see from the table, **except for the first class with largest label frequency, our method has promoted the performance of all remaining classes**. Besides, We also conducted an analysis of the FID scores for each class under a different assignment of head and tail categories. **The detailed information could be found in Appendix. F**.
>
> ### W3 & Q4
> > The author lacks experimental validation on large-scale datasets.
>
> > How effective are the proposed methods on larger datasets (ImageNet-LT)?
>
> We have added one experiments on ImageNet-LT dataset. The following table lists the results. Our results on large-scale are consistent to the observations on small-scale data, which validates the effectiveness of our method at scale.
>
> | Method    | FID &thinsp;$\downarrow$  | IS &thinsp;$\uparrow$    |
> |:---------:|:-----:|:------:|
> | base DDPM | 26.95 | **15.99**  |
> | CBDM      | 26.12 | 15.86  |
> | T2H(Ours) | **25.42** | 15.96  |
>
> We have included this experiment and presented more discussion in Appendix. I.

---

### Author Response · Authors · 2023-11-18
**General Response by Authors**

We extend our sincere gratitude to the reviewers for their insightful feedback. We deeply appreciate the substantial efforts invested by all the reviewers, Area Chairs (AC), and Program Chairs (PC) in reviewing our work. Their contributions have been invaluable in enhancing the quality of our research.

According to the suggestions, we have carefully revised and modified the paper in response to the comments and concerns from all reviewers. In addition to provide a detailed response to each reviewer, here we summary the main point:

- We restate the importance of Batch Re-sample, a simple yet direct method. On one hand, it addresses the issue of **head class score dominance mainly in unconditional generation, and on the other, it amplifies the strength of head-to-tail transfer via alleviating the limitation on overall transfer strength in T2H mode under conditional generation**. Because Batch Re-sample is a commonly used method in long tail recognition, we slightly abbreviated the method description and give the credit to the previous works in the paper.
- We have **reorganized the structure of the 'Method' section for enhanced clarity and readability**. The placement of Proposition 3.1 in the article was adjusted to clarify the motivation of using Batch Resampling with T2H.
- In order to elucidate the motivation of T2H more effectively, We introduced T2H from a more intuitive and fundamental perspective and incorporated a new figure (Figure.3) for better understanding.

- We have included experiments on ImageNet-LT to further substantiate the effectiveness of our method on large-scale data.

---

### Author Response · Authors · 2023-11-23
**Gratitude to Reviewers and Further Paper Enhancements**

Thanks to all reviewers for their valuable advice. We have made improvements to both the content, including the abstract and main text, and the clarity of our figures to provide a more comprehensive and illustrative presentation. Once again, we extend our appreciation to all the reviewers for their peer review, constructive feedback, and contributions that have enhanced the quality of our paper.

---

### Meta-Review · Area_Chair_Nfws · 2023-12-05

**Metareview:**

This paper considers the problem of improving the synthesis quality of diffusion models on long tail categories. While the current diffusion models can perform well on common classes, they still underperform on rare classes, which makes this problem an important one to address. The authors rewrite the target for the score estimation as mixture of multiple targets computed from a reference batch, where a balanced reference batch is sampled using a ”Batch Resample” strategy. The authors also propose T2H and H2T strategies for further calibrating the scores. Experiments on small-scale datasets show improved performance.

The main concerns raised by reviewers include limited novelty, poor presentation and lack of experiments on large scale datasets. Regarding novelty, while "Batch Resample" is a commonly used technique, the authors are the first to show it in the context of diffusion models. I believe this is a worthwhile contribution. In the rebuttal, the authors have improved presentation in several sections, notably in the proofs and Section 3. The reviewers agree that the presentation has been improved in the revised draft. Regarding experiments, I agree that large scale experiments are missing. However, the results on small scale datasets are good. In particular, the performance on each category shown in rebuttal makes it very clear that the performance on tail categories improves.

Taking all these into consideration, I think the paper has sufficient contribution. All reviewers are leaning towards acceptance as well. So, I vote for accepting this paper.

**Justification For Why Not Higher Score:**

While the paper has decent contribution, the novelty is somewhat limited and not up to the mark of a Oral paper. The experimental validation can be improved with more analysis.

**Justification For Why Not Lower Score:**

The paper has addressed an important problem and shown good results on small-scale experiments. I believe the contribution is sufficient for an accept with poster.

---

### Decision · Program_Chairs · 2024-01-16

Accept (poster)